# BAZ2A-RNA mediated association with TOP2A and KDM1A represses genes implicated in prostate cancer

Marcin Roganowicz[1,2], Dominik Bär[1], Cristiana Bersaglieri[1] ⓘ, Rossana Aprigliano[1], Raffaella Santoro[1] ⓘ

BAZ2A represses rRNA genes (rDNA) that are transcribed by RNA polymerase I. In prostate cancer (PCa), BAZ2A function goes beyond this role because it represses genes frequently silenced in metastatic disease. However, the mechanisms of this BAZ2A-mediated repression remain elusive. Here, we show that BAZ2A represses genes through its RNA-binding TAM domain using mechanisms differing from rDNA silencing. Although the TAM domain mediates BAZ2A recruitment to rDNA, in PCa, this is not required for BAZ2A association with target genes. Instead, the BAZ2A-TAM domain in association with RNA mediates the interaction with topoisomerase 2A (TOP2A) and histone demethylase KDM1A, whose expression positively correlates with BAZ2A levels in localized and metastatic PCa. TOP2A and KDM1A pharmacological inhibition up-regulate BAZ2A-repressed genes that are regulated by inactive enhancers bound by BAZ2A, whereas rRNA genes are not affected. Our findings showed a novel RNA-based mechanism of gene regulation in PCa. Furthermore, we determined that RNA-mediated interactions between BAZ2A and TOP2A and KDM1A repress genes critical to PCa and may prove to be useful to stratify prostate cancer risk and treatment in patients.

## Introduction

Prostate cancer (PCa) is the second most frequent cancer in men and the fifth leading cause of cancer death (Sung et al, 2021). Most of the PCa are indolent ones, with no threat to mortality; however, in many cases, an aggressive form of disease develops and progresses to metastasis, which accounts for almost two thirds of PCa-related deaths (Siegel et al, 2015). PCa displays high heterogeneity that leads to distinct histopathological and molecular features (Li & Shen, 2018). This heterogeneity posits one of the most confounding and complex factors underlying its diagnosis, prognosis, and treatment (Yadav et al, 2018).

BAZ2A (also known as TIP5) is a chromatin repressor known for its role in the silencing of rRNA genes, which are transcribed by the RNA polymerase I (Pol I) (Santoro et al, 2002; Guetg et al, 2010). Recent studies showed that BAZ2A is also implicated in several cancers such as PCa and hepatocellular carcinoma (Gu et al, 2015; Li et al, 2018). In the case of PCa, it was shown that the BAZ2A function goes beyond the regulation of rRNA genes and is implicated in aggressive diseases (Gu et al, 2015; Pietrzak et al, 2020; Peña-Hernández et al, 2021). BAZ2A is highly expressed in metastatic PCa and involved in maintaining PCa cell growth by repressing genes frequently silenced in metastatic PCa (Gu et al, 2015; Peña-Hernández et al, 2021). Moreover, elevated BAZ2A level indicates poor outcome (Pietrzak et al, 2020) and is also of high prognostic value for predicting PCa recurrence (Gu et al, 2015). However, the mechanisms of how BAZ2A mediates gene repression in PCa remain yet elusive.

BAZ2A contains a RNA-binding domain known as TAM (TIP5/ARBD/MBD). The TAM domain plays a key role in rRNA gene silencing through the interaction with the long noncoding (lnc) promoter (p)RNA, a transcript originating from an alternative promoter of the rRNA genes (Mayer et al, 2006; Guetg et al, 2012; Anosova et al, 2015). The association of BAZ2A-TAM domain with pRNA is required for the interaction with the transcription termination factor I (TTF1) that mediates BAZ2A recruitment to rRNA genes and transcriptional repression (Savić et al, 2014; Leone et al, 2017). However, it remains yet to elucidate whether BAZ2A uses similar RNA-mediated mechanisms to repress genes implicated in aggressive PCa.

In this work, we show that BAZ2A represses genes in PCa through its RNA-binding TAM domain using mechanisms that substantially differ from the ones used to silence rRNA genes. We found that rRNA gene expression in PCa cells is still affected by pRNA and BAZ2A, indicating that PCa cells retain similar mechanisms to establish rRNA gene silencing that were described in non-cancer cells (Mayer et al, 2006; Guetg et al, 2010; Leone et al, 2017). In contrast, the other BAZ2A-regulated genes in PCa do not depend on pRNA. Furthermore, BAZ2A-TAM domain is not required for the recruitment to its target genes. Instead, TAM domain mediates BAZ2A association with factors implicated in the regulation of gene expression, chromatin organization, nuclear pore complex, and RNA splicing. Among BAZ2A-TAM-dependent interacting proteins,

[1]Department of Molecular Mechanisms of Disease, DMMD, University of Zurich, Zurich, Switzerland   [2]RNA Biology Program, Life Science Zurich Graduate School, University of Zurich, Zurich, Switzerland

Correspondence: raffaella.santoro@dmmd.uzh.ch

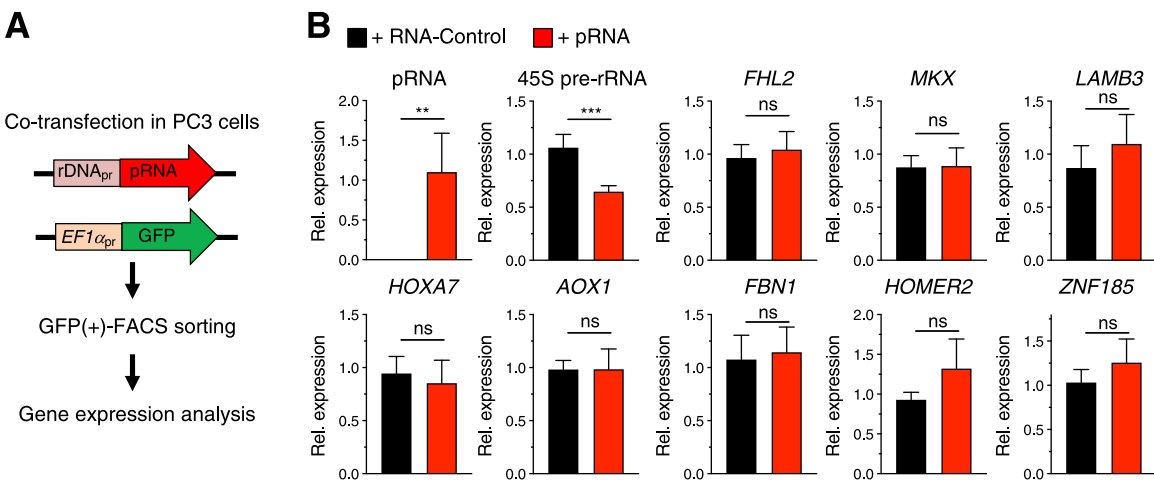

**Figure 1. BAZ2A-mediated silencing of PCa target genes does not depend on pRNA.**
**(A)** A scheme of the strategy used to measure gene expression in PC3 cells overexpressing the lncRNA pRNA. rDNA$_{pr}$ represents the promoter of human rRNA genes. GFP expression was under the control of EF1α promoter. **(B)** qRT–PCR of GFP(+)-FACS-sorted PC3 cells showing pRNA, 45S pre-rRNA, and mRNA levels of known and validated BAZ2A target genes that are repressed by BAZ2A (Gu et al, 2015). Transcript levels were normalized to *GAPDH* and cells transfected with RNA-control. Error bars represent SD of three independent experiments. Statistical significance (*P*-value) was calculated using two-tailed *t* test (**<0.01, ***< 0.001); ns, not significant. Source data are available for this figure.

we found topoisomerase 2A (TOP2A) and histone demethylase KDM1A, which have been previously implicated in aggressive PCa (Labbé et al, 2017; Liang et al, 2017; Sehrawat et al, 2018). Importantly, BAZ2A-TAM mediated interaction with TOP2A and KDM1A requires RNA. TOP2A and KDM1A expression levels positively correlate with BAZ2A levels in both localized and aggressive PCa. Moreover, the repression of BAZ2A-TAM-dependent genes correlates with primary and metastatic tumours with high BAZ2A, KDM1A, and TOP2A content. Finally, pharmacological inhibition of TOP2A and KDM1A activity specifically up-regulates the expression of BAZ2A-TAM-repressed genes that are regulated by a class of inactive enhancers bound by BAZ2A. In contrast, the expression of rRNA genes was not affected by TOP2A and KDM1A inhibition, highlighting different BAZ2A-mediated repressive mechanisms that depend on the gene context. Our findings showed a novel RNA-based mechanism of gene regulation in PCa. Furthermore, the data indicate that RNA-mediated interactions between BAZ2A and TOP2A and KDM1A regulate genes critical to PCa and may prove to be useful for the stratification of PCa risk and treatment in patients.

## Results

### BAZ2A-mediated silencing of PCa target genes does not depend on pRNA

Previous results showed that the function of BAZ2A in PCa goes beyond the silencing of rRNA genes because it can also repress other genes that are linked to aggressive PCa (Gu et al, 2015; Peña-Hernández et al, 2021). BAZ2A is an RNA-binding protein through its TAM domain (Mayer et al, 2006; Guetg et al, 2012; Savić et al, 2014; Anosova et al, 2015). In BAZ2A-mediated rRNA gene silencing, the BAZ2A-TAM domain plays an important role because its interaction with the lncRNA pRNA is required for BAZ2A recruitment to the

promoter of rRNA genes (Mayer et al, 2006; Savić et al, 2014; Leone et al, 2017). To determine whether in PCa cells BAZ2A represses genes other than rRNA genes using similar mechanisms required for the establishment of rRNA gene silencing, we asked whether pRNA could affect the expression of known BAZ2A-repressed genes in PCa cells (Gu et al, 2015). We measured the expression of validated BAZ2A target genes that were found in our previous studies to be bound and repressed by BAZ2A in PC3 cells and frequently silenced in metastatic PCa (Gu et al, 2015; Pena-Hernandez et al, 2021). To determine the role of pRNA for the regulation of these BAZ2A-repressed genes, we co-transfected PC3 cells with constructs expressing RNA-control, a sequence previously shown to have low binding affinity to BAZ2A, or pRNA which strongly interacts with BAZ2A (Guetg et al, 2012; Savić et al, 2014). Both RNAs have similar sizes (ca. 200 nt) and their expressions were put under the control of the human rRNA gene promoter that allows transcription driven by RNA polymerase I, which transcribes pRNA in cells. These same constructs also contained GFP under the control of EF-1α promoter, thereby allowing the isolation of GFP(+)-cells by FACS sorting 72 h posttransfection (Fig 1A). Consistent with previous results, the overexpression of pRNA caused down-regulation of 45S pre-rRNA, indicating that pRNA down-regulates rRNA gene expression in PCa cells as well (Mayer et al, 2006) (Fig 1B). In contrast, pRNA did not cause significant changes in the expression of the other known BAZ2A-repressed genes (Gu et al, 2015), suggesting that their regulation is pRNA independent. These results suggest that BAZ2A represses these genes using mechanisms that differ from the silencing of rRNA genes that requires pRNA.

### BAZ2A regulates a set of genes involved in PCa through its TAM domain

To determine whether BAZ2A-mediated gene regulation in PCa cells depends on the TAM domain, we performed RNA-seq analysis on

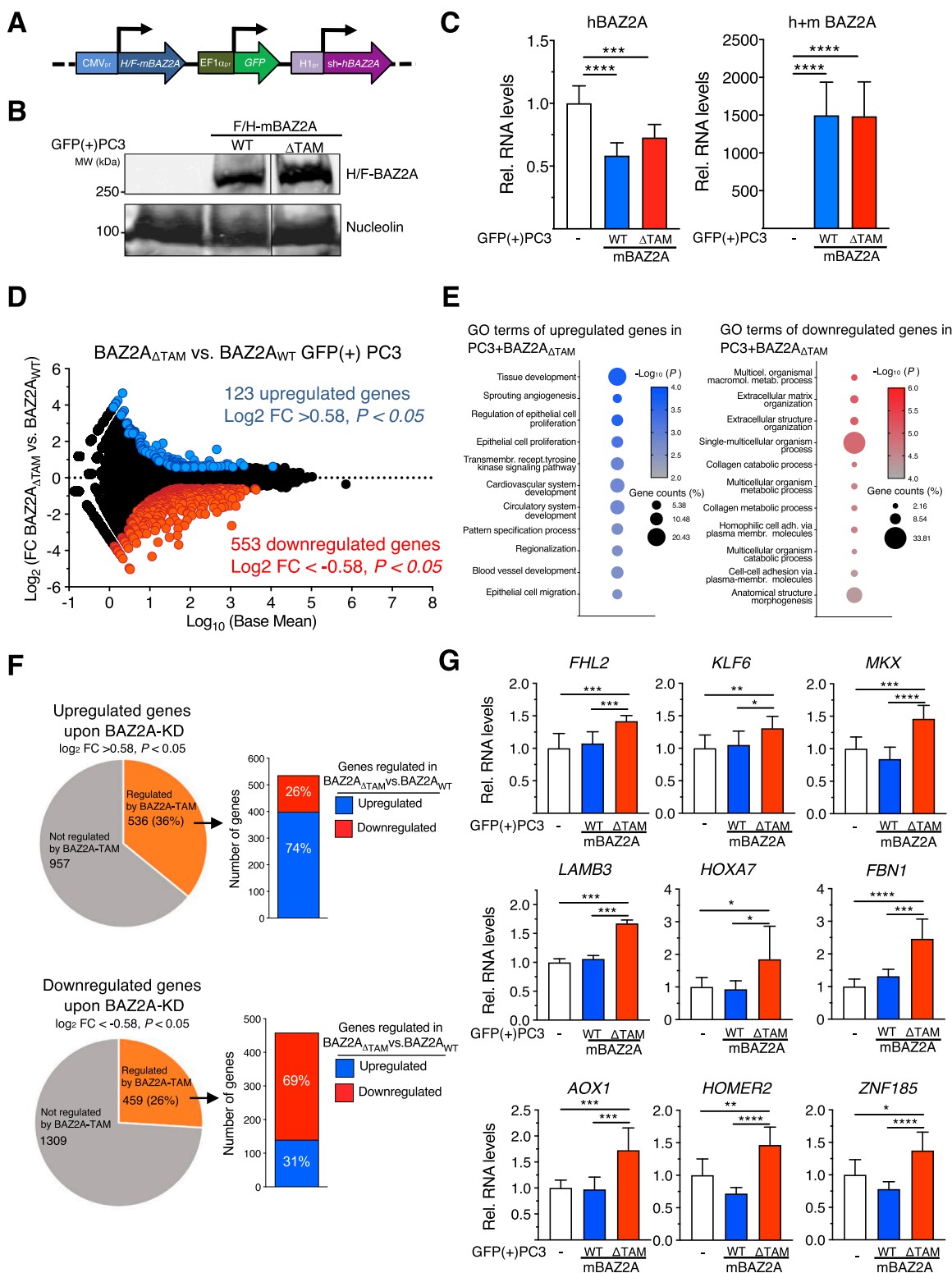

**Figure 2. BAZ2A regulates a set of genes involved in PCa development through its RNA-binding TAM domain.**
**(A)** A scheme of the all-in-one plasmids for the expression of mouse HA-FLAG BAZ2A (F/H-mBAZ2A) under CMV promoter, GFP reporter gene under EF1α promoter, and shRNA for endogenous human (h)BAZ2A under H1 promoter. **(B)** Western blot of GFP(+)-FACS-sorted PC3 cells transfected with all-in-one plasmids expressing F/H-mBAZ2A$_{WT}$ and F/H-mBAZ2A$_{ΔTAM}$. The BAZ2A signal was detected with anti-HA antibodies. Nucleolin served as a loading control. The line indicates that the blot is not continuous. **(C)** GFP(+)-FACS-sorted PC3 cells transfected with all-in-one constructs expressing F/H-mBAZ2A$_{WT}$ and F/H-mBAZ2A$_{ΔTAM}$ with simultaneous down-regulation of endogenous human BAZ2A. qRT–PCR showing mRNA levels of human BAZ2A (left) and total (human and mouse) BAZ2A (right). Control cells were transfected

PC3 cells expressing BAZ2A$_{WT}$ and BAZ2A$_{\Delta TAM}$. We engineered all-in-one constructs that express F/H-tagged mouse BAZ2A$_{WT}$ (F/H-mBAZ2A$_{WT}$) or BAZ2A$_{\Delta TAM}$ (F/H-mBAZ2A$_{\Delta TAM}$) under CMV promoter, the GFP reporter gene under EF-1α promoter, and shRNA against human (h)BAZ2A sequences that do not target m*BAZ2A* (Fig 2A). hBAZ2A and mBAZ2A share high sequence homology (86%) and identical TAM domain sequences. The expression of shRNA that specifically targets hBAZ2A was reasoned to obtain the expression of ectopic mBAZ2A$_{WT}$ and mBAZ2A$_{\Delta TAM}$, allowing simultaneous down-regulation of the endogenous hBAZ2A. We transfected PC3 cells with the all-in-one constructs and isolated GFP(+)-cells by FACS sorting 72 h later. We confirmed equal expression of mBAZ2A$_{WT}$ and mBAZ2A$_{\Delta TAM}$ by Western blot and qRT–PCR (Fig 2B and C) and the reduction of endogenous hBAZ2A in FACS-sorted GFP(+) PC3 cells (Fig 2C). As expected, because of transient transfection, the expression of ectopic mBAZ2A was higher than endogenous hBAZ2A. Correlation analysis clustered separately PCa cells expressing mBAZ2A$_{\Delta TAM}$ or mBAZ2A$_{WT}$, indicating a differential gene expression profile (Fig S1). We identified 123 up-regulated genes and 553 down-regulated genes in PC3 cells expressing mBAZ2A$_{\Delta TAM}$ compared with cells expressing mBAZ2A$_{WT}$ (log$_2$ fold change ± 0.58, *P* < 0.05, Fig 2D and Table S1). The top GO terms for the up-regulated genes were related to tissue and organ development, cell migration and locomotion, regulation of epithelial cells proliferation, and angiogenesis (Fig 2E and Table S2). Many of these processes are recognized as hallmarks of cancer, such as dedifferentiation, proliferative signalling, angiogenesis induction, and activation of invasion and metastasis (Hanahan & Weinberg, 2011; Zhong et al, 2020). The GO terms for down-regulated genes were mostly related to metabolic processes that are also known to contribute to various cancers' development by fuelling cancer cells' growth and division (Hanahan & Weinberg, 2011).

Next, we used published RNA-seq data (Peña-Hernández et al, 2021) to define differentially expressed genes in PC3 cells depleted of BAZ2A by siRNA (BAZ2A-regulated genes; log$_2$ fold change ±0.58, *P* < 0.05) and asked how many BAZ2A-regulated genes were significantly affected by BAZ2A$_{\Delta TAM}$ expression (Fig 2F). We found that 36% and 26% of up-regulated and down-regulated genes upon BAZ2A-knockdown (KD), respectively, were significantly affected by BAZ2A$_{\Delta TAM}$ expression. Remarkably, most of these BAZ2A-regulated genes showed similar changes in gene expression upon the expression of BAZ2A$_{\Delta TAM}$. A large portion (74%) of these genes up-regulated upon BAZ2A-KD were significantly up-regulated upon the expression of BAZ2A$_{\Delta TAM}$. Similarly, 69% of genes down-regulated upon BAZ2A-KD were also down-regulated by BAZ2A$_{\Delta TAM}$. These

results suggest a major role of the BAZ2A-TAM domain in affecting BAZ2A-regulated gene expression in PCa cells. We further supported these results by measuring through qRT–PCR the expression of validated BAZ2A-repressed genes in PCa (Gu et al, 2015) that did not depend on pRNA (Fig 1D) and found that they were significantly up-regulated in cells expressing BAZ2A$_{\Delta TAM}$ compared with PC3 cells expressing BAZ2A$_{WT}$ or only GFP (Fig 2G). These results indicated that the repression of these BAZ2A-repressed genes is dependent on the BAZ2A-TAM domain but pRNA independent. We did not observe any evident additional down-regulation upon mBAZ2A$_{WT}$ expression, suggesting that these genes are already repressed by BAZ2A and cannot be further down-regulated by increasing BAZ2A expression levels. Taken together, these results point for a role of BAZ2A in the regulation of the expression of genes implicated in cancer-related processes through its TAM domain.

## BAZ2A-TAM domain is required for the repression of a set of genes that have their enhancers bound by BAZ2A

Previous BAZ2A-ChIPseq analyses in PC3 cells determined that BAZ2A associates with a class of inactive enhancers that are enriched in histone H3 acetylated at lysin 14 (H3K14ac) and depleted of H3K27ac and H3K4me1 (Peña-Hernández et al, 2021) (Fig 3A). This group of inactive enhancers was termed class 2 (C2) enhancers and appears to be the major mechanism of BAZ2A-regulation in PCa cells because only few BAZ2A-regulated genes had their promoter associated with BAZ2A (Peña-Hernández et al, 2021). The interaction of BAZ2A with C2-enhancers was shown to be mediated by the BAZ2A-bromodomain that specifically binds to H3K14ac. BAZ2A-bound C2-enhancers are significantly enriched in H3K14ac and the repressive mark H3K27me3 and depleted of the active modification H3K27ac relative to the active enhancer class C1 marked by H3K27ac, H3K4me1, and DNase I hypersensitive sites (Peña-Hernández et al, 2021) and C2 enhancers not bound by BAZ2A (Fig 3B). Importantly, genes in the nearest linear proximity to BAZ2A-bound C2 enhancers were found to be repressed by BAZ2A, indicating that BAZ2A could repress genes through the association with this class of inactive enhancers. To determine whether BAZ2A-TAM domain is required for the repression of these genes, we analysed the RNA-seq profiles of PC3 cells expressing BAZ2A$_{WT}$ and BAZ2A$_{\Delta TAM}$ and found that 25% (240) of the genes in the nearest linear proximity to BAZ2A-bound C2-enhancers were significantly affected by BAZ2A$_{\Delta TAM}$ compared with BAZ2A$_{WT}$ (Fig 3A, C, and D). Remarkably, most of these BAZ2A-TAM-regulated genes (202, 84%) were significantly up-regulated upon the expression of BAZ2A$_{\Delta TAM}$,

with plasmids only expressing GFP. mRNA levels were normalized to *GAPDH*. Error bars represent SD of three independent experiments. Statistical significance (*P*-value) was calculated using paired two-tailed *t* test (***< 0.001, ****<0.0001). **(D)** BAZ2A regulates gene expression in PC3 cells. MA plot of the log$_2$ fold change (FC) gene expression of GFP(+)-FACS-sorted PC3 cells expressing BAZ2A$_{\Delta TAM}$ compared with cells expressing BAZ2A$_{WT}$. Gene expression values of three replicates were averaged. Blue points represent up-regulated genes (log$_2$FC > 0.58) and red points show down-regulated genes (log$_2$FC < −0.58) and *P*-value < 0.05. **(E)** Top 11 gene ontology (GO) terms as determined using DAVID for genes up-regulated and down-regulated in PC3 cells expressing BAZ2A$_{\Delta TAM}$ compared with cells expressing BAZ2A$_{WT}$. **(F)** Pie chart showing the number of genes significantly affected by BAZ2A$_{\Delta TAM}$ expression that are up-regulated (upper panel) or down-regulated (bottom panel) in PC3 cells upon Baz2A knockdown (Pena-Hernandez et al, 2021). On the right, it is shown the number of genes up-regulated and down-regulated upon BAZ2A$_{\Delta TAM}$ expression. **(G)** BAZ2A regulates the expression of genes frequently repressed in metastatic mPCa through its TAM domain. qRT–PCR showing mRNA levels of the genes frequently repressed in mPCa upon expression of BAZ2A$_{WT}$ and BAZ2A$_{\Delta TAM}$ domains in GFP(+)-FACS-sorted PC3 cells. Control cells were transfected with plasmids only expressing GFP. mRNA levels were normalized to *GAPDH* mRNA and control cells transfected with plasmid-expressing GFP. Error bars represent SD of three independent experiments. Statistical significance (*P*-value) was calculated using two-tailed *t* test (*<0.05, **<0.01, ***< 0.001, ****<0.0001). Source data are available for this figure.

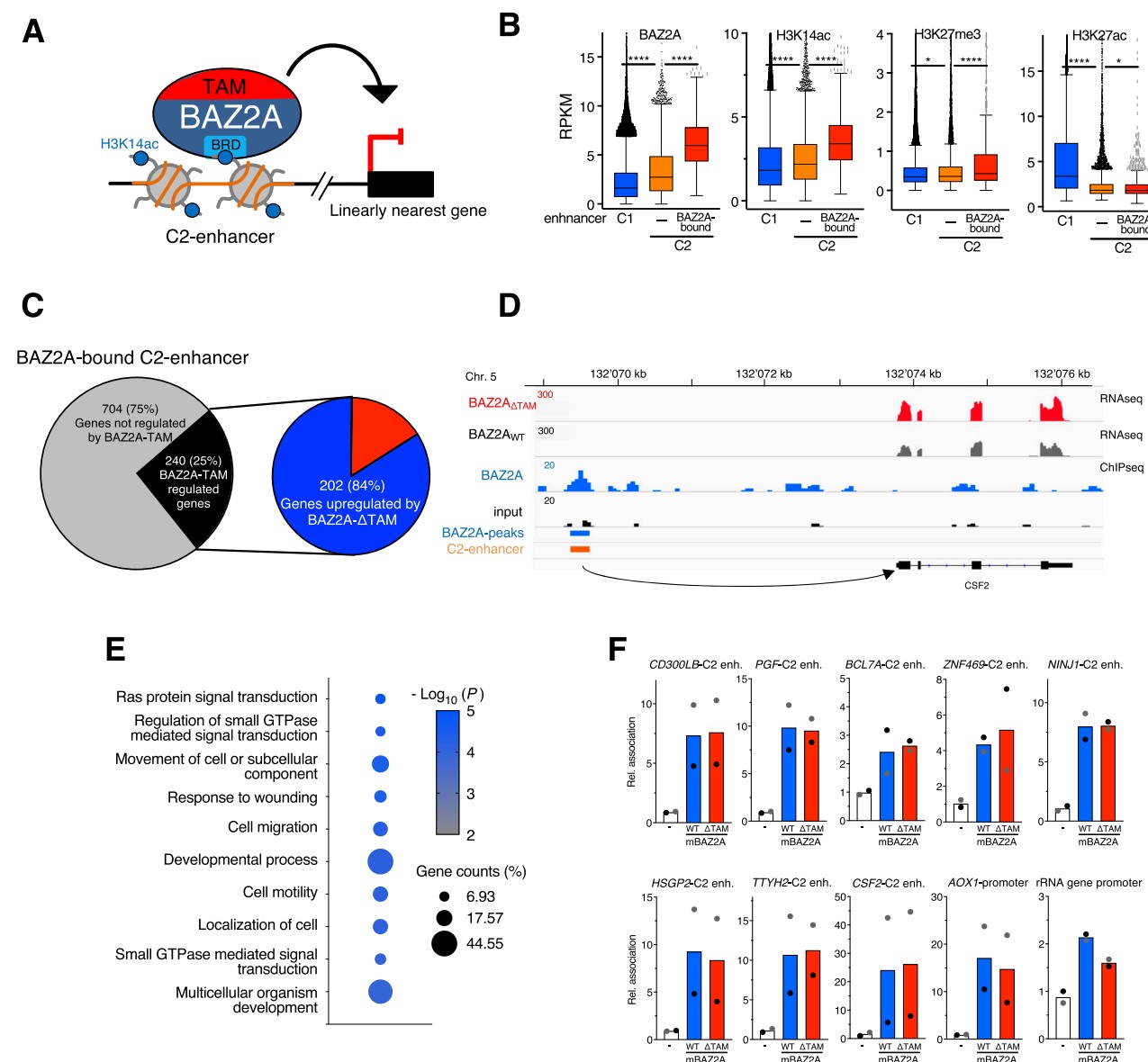

**Figure 3. BAZ2A-TAM domain is required for the repression of a set of genes that have their enhancers bound by BAZ2A.**
**(A)** A schema showing BAZ2A-BRD-mediated association with C2-enhancer-enriched in H3K14ac and the linearly nearest gene in PC3 cells. **(B)** Levels of BAZ2A and the histone modifications H3K14ac, H3K27me3, and H3K27ac at active C1-enhancers (i.e., active enhancers marked by H3K27ac, H3K4me1, and DNase I hypersensitive sites) and inactive C2-enhancer bound or not bound by BAZ2A (Peña-Hernández et al, 2021). Read coverage was measured at ±1 Kb from annotated enhancer regions. Values are shown as average RPKM. Statistical significance (*P*-values) was calculated using the unpaired two-tailed *t* test (*<0.05, ****<0.0001). **(C)** Right panel. Pie charts showing the number of genes in the nearest linear genome proximity to BAZ2A-bound C2-enhancers and their significant expression changes in PC3 cells expressing BAZ2A$_{\Delta TAM}$ compared with PC3 cells expressing BAZ2A$_{WT}$. Genes significantly affected by BAZ2A$_{\Delta TAM}$ expression: log$_2$ fold change ±0.1, *P* < 0.05. **(D)** Wiggle tracks showing RNA levels in PC3 cells expressing BAZ2A$_{WT}$ and BAZ2A$_{\Delta TAM}$, occupancy of BAZ2A, C2 enhancer, and its nearest gene *CSF2*. **(E)** Top 10 gene ontology (GO) terms of genes in the nearest linear genome proximity to BAZ2A-bound C2-enhancers and significantly up-regulated in PC3 cells expressing BAZ2A$_{\Delta TAM}$. **(F)** BAZ2A-TAM domain is not implicated in BAZ2A association with C2-enhancers. HA-ChIP analysis of PC3 cells transfected with vectors expressing only GFP (−), F/H- BAZ2A$_{WT}$, and F/H-BAZ2A$_{\Delta TAM}$ showing the association of BAZ2A with the nearest C2-enhancers (enh.) to genes derepressed by BAZ2A$_{\Delta TAM}$. The association with the promoter of BAZ2A-regulated gene *AOX1* and rRNA genes is also shown. Values were normalized to the corresponding inputs and control samples (−). Values are from two independent experiments. Grey and black dots represent the values obtained from each single experiment.

suggesting a major role of BAZ2A-TAM domain in repressing gene transcription through its interaction with C2-enhancers. These results also suggest that genes down-regulated upon the expression of BAZ2A$_{\Delta TAM}$ (Fig 2E) might not be direct targets of BAZ2A and are probably affected through indirect mechanisms. The

top GO terms of genes repressed through BAZ2A-TAM domain showed a high proportion of them (44.55%) significantly linked to developmental process, suggesting that BAZ2A promotes cell de-differentiation, a mark of aggressive PCa (Han et al, 2022) (Fig 3E and Table S3). Interestingly, the analysis of a comprehensive dataset

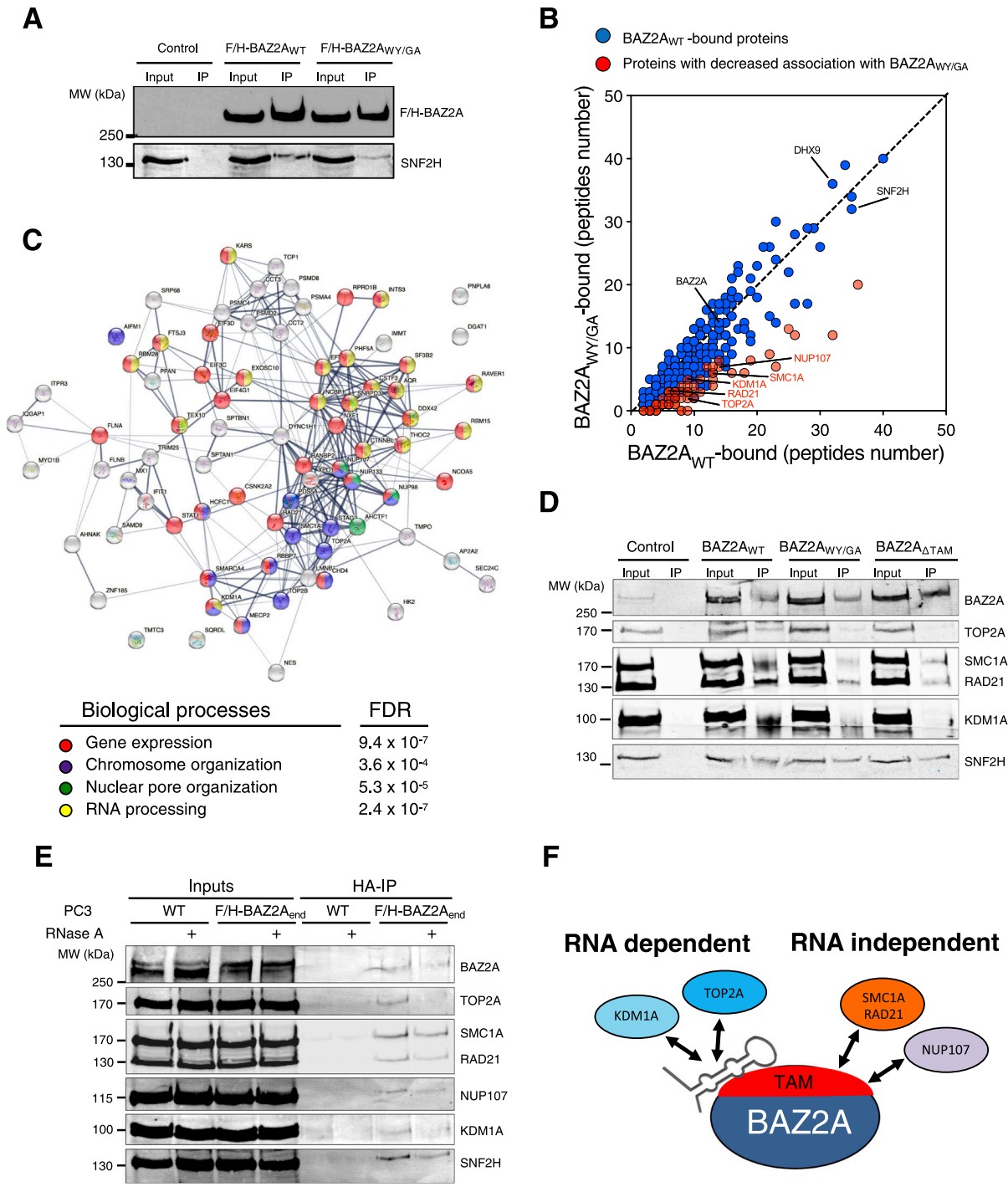

**Figure 4. BAZ2A associates with TOP2A and KDM1A through TAM domain and RNA in PCa cells.**
**(A)** HA-BAZ2A immunoprecipitation from nuclear extracts of PC3 cells transfected with plasmids expressing F/H-BAZ2A$_{WT}$ and F/H-BAZ2A$_{WY/GA}$. The BAZ2A signal was detected with anti-HA antibodies. The interaction with SNF2H was visualized with anti-SNF2H antibodies. **(B)** Mass spectrometry analysis of F/H-BAZ2A$_{WT}$ and F/H-BAZ2A$_{WY/GA}$ immunoprecipitates from PC3 cells. Values represent average of peptide numbers from three independent experiments. Values in orange represent proteins with decreased association with BAZ2A$_{WY/GA}$ compared with BAZ2A$_{WT}$. **(C)** STRING analysis depicting functional protein association networks of BAZ2A-interacting proteins that depend on a functional TAM domain. **(D)** HA-BAZ2A immunoprecipitation of nuclear extracts for PC3 cells transfected with plasmids expressing F/H-BAZ2A$_{WT}$, F/H-BAZ2A$_{WY/GA}$, and F/H-BAZ2A$_{\Delta TAM}$. BAZ2A-interacting proteins are visualized in immunoblots with the corresponding antibodies. **(E)** TOP2A and KDM1A interactions with BAZ2A depend on RNA. HA-immunoprecipitation from nuclear extract of the PC3 cell line expressing endogenous BAZ2A with F/H tag (F/H-BAZ2Aend) and parental PC3 cells that were treated with or without RNase A (0.1 μg/μl). BAZ2A-interacting proteins are visualized in immunoblots with the corresponding antibodies. **(F)** Schema representing the role of BAZ2A-TAM domain and RNA in mediating BAZ2A association with TOP2A, KDM1A, SMC1A, RAD21, and NUP107.
Source data are available for this figure.

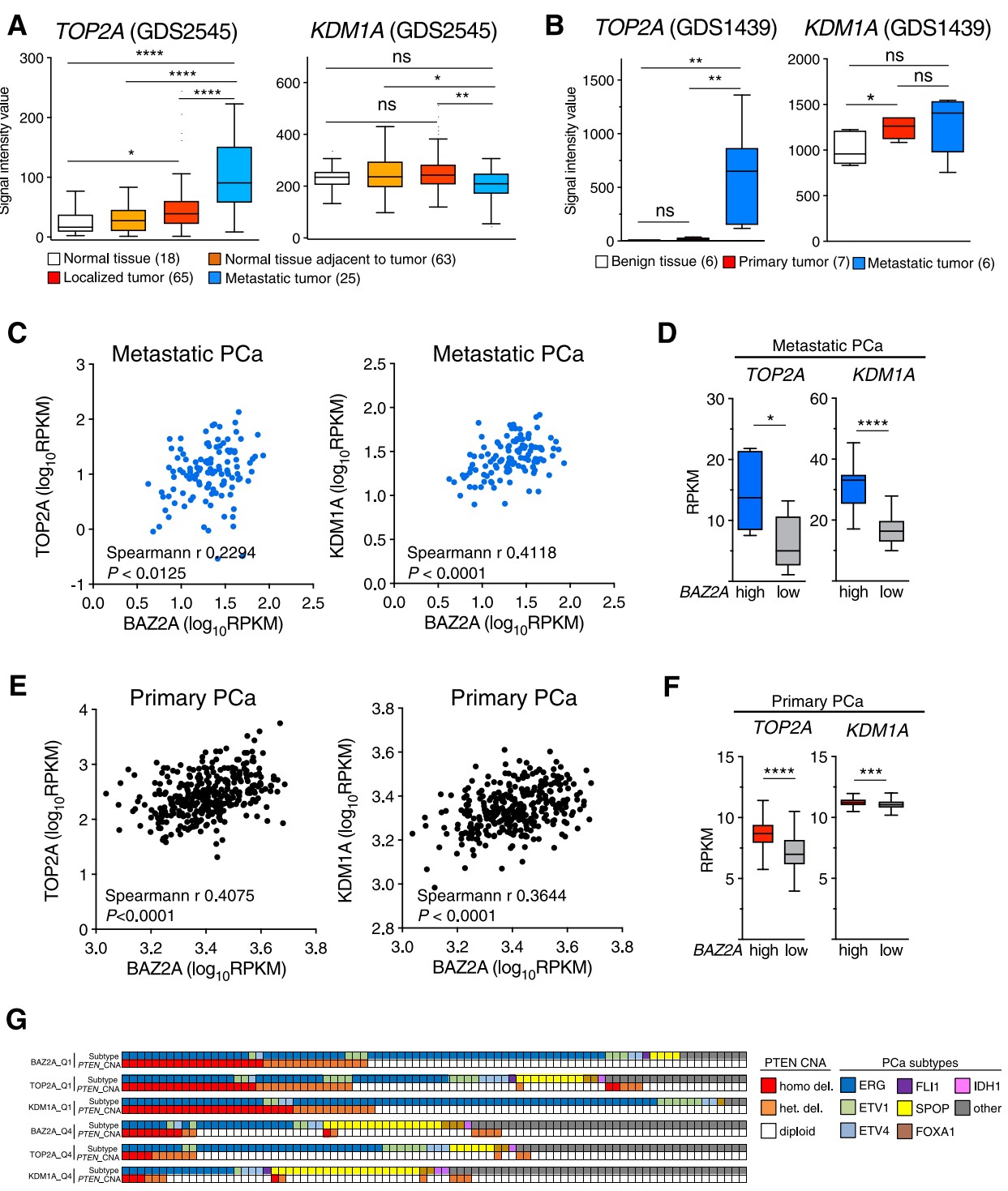

**Figure 5. TOP2A and KDM1A expression levels correlate with BAZ2A levels in primary and metastatic PCa.**
**(A, B)** Boxplots showing *TOP2A* (A) and *KDM1A* (B) transcript levels from gene expression microarray GEO datasets GDS2545 (Yu et al, 2004; Chandran et al, 2007) and GDS1439 (Varambally et al, 2005). Statistical significance (*P*-value) was calculated using two-tailed *t* test (* <0.05, **<0.01, ****<0.0001); ns, not significant. **(C)** Scatter plot showing *BAZ2A* levels relative to *TOP2A* (left panel) and *KDM1A* (right panel) levels from metastatic prostate tumours. Dataset is from Robinson et al (2015). Statistical significance was calculated using Spearmann correlation. **(D)** Box plot showing *TOP2A* (left panel) and *KDM1A* (right panel) levels in metastatic tumours expressing high or low BAZ2A levels. Quartile 1: the top 25% of PCas with the highest BAZ2A expression. Quartile 4: the top 25% of PCas with the lowest BAZ2A levels. Error bars represent s.d. Statistical significance (*P*-values) for the experiments was calculated using two-tailed *t* test (*** < 0.001). **(E)** Scatter plot showing *BAZ2A* levels relative to *TOP2A* (left panel) and *KDM1A* (right panel) levels from primary prostate tumours. The dataset is from Cancer Genome Atlas Research Network (2015). Statistical significance was calculated using Spearmann correlation. **(F)** Box plot showing *TOP2A* (left panel) and *KDM1A* (right panel) levels in primary tumours expressing high or low BAZ2A levels. Quartile 1: the top 25% of PCas with the highest BAZ2A expression. Quartile 4: the top 25% of PCas with the lowest BAZ2A levels. Error bars represent s.d. Statistical significance

comparing gene expression between metastatic tumours to normal tissue (Yu et al, 2004; Chandran et al, 2007) showed that about 40% of genes up-regulated by BAZ2A$_{\Delta TAM}$ and linked to developmental processes are repressed in metastatic PCa (Table S3 and Fig S2A and B).

Because the TAM domain was shown to be required for BAZ2A association with rRNA genes (Mayer et al, 2006), we asked whether this was also the case for BAZ2A interaction with C2-enhancers. We performed ChIP analysis of PC3 cells transfected with F/H-mBAZ2A$_{WT}$ or F/H-mBAZ2A$_{\Delta TAM}$ and analysed BAZ2A association with BAZ2A-bound C2-enhancers that were close to genes up-regulated by BAZ2A$_{\Delta TAM}$. However, we did not find any significant alterations in the binding of BAZ2A$_{\Delta TAM}$ with these regions relative to BAZ2A$_{WT}$ (Fig 3F). Similarly, the interaction with the promoter of *AOX1*, a known gene directly regulated by BAZ2A (Gu et al, 2015; Peña-Hernández et al, 2021), was not affected. These results indicate that the interaction with these loci is not mediated by BAZ2A-TAM domain. Furthermore, they support previous data showing that it is the BAZ2A-BRD domain to be required for the interaction of C2-enhancers that contain H3K14ac (Peña-Hernández et al, 2021). Consistent with previous results (Mayer et al, 2006), the ChIP analysis showed that the association of BAZ2A$_{\Delta TAM}$ with the promoter of rRNA genes was reduced compared with BAZ2A$_{WT}$ (Fig 3F). Thus, although for rRNA gene silencing, BAZ2A-TAM is required for the association with chromatin, in the case of the other BAZ2A-regulated genes, BAZ2A-TAM domain is not necessary for the recruitment to target loci, suggesting different mechanisms by which BAZ2A represses gene expression in PCa cells.

### BAZ2A associates with KDM1A and TOP2A in a TAM domain and RNA-dependent manner

We hypothesized that BAZ2A-TAM domain and its ability to interact with RNA might be important for the association with factors implicated in BAZ2A-mediated gene repression in PCa cells. To test this, we performed HA-IP combined with mass spec analysis from PC3 cells transfected with plasmids expressing F/H-BAZ2A$_{WT}$ or the deficient RNA-binding mutant F/H-BAZ2A$_{WY/GA}$. In this series of experiments, we reasoned to use BAZ2A$_{WY/GA}$ instead of BAZ2A$_{\Delta TAM}$ mutant to obtain a more specific identification of proteins requiring RNA to interact with BAZ2A. The IP specificity was validated by Western blot with antibodies against HA to detect F/H-BAZ2A and against SNF2H, a well-known BAZ2A-interacting protein (Strohner et al, 2001; Dalcher et al, 2020) (Fig 4A). We defined as BAZ2A$_{WT}$-interacting proteins those factors showing in all the three replicates at least two peptides and ≥twofold peptide number in IPs with F/H-BAZ2A$_{WT}$ relative to IPs from cells transfected with an empty vector (Table S4). To identify factors that specifically associate with BAZ2A through a functional BAZ2A-TAM domain, we considered BAZ2A$_{WT}$-interacting proteins that showed ≥twofold peptide number in BAZ2A$_{WT}$-IPs relative to BAZ2A$_{WY/GA}$-IPs in at least two independent

replicates (Fig 4B and Table S4). Using these criteria, proteins previously shown to interact with BAZ2A independently of RNA, such as SNF2H and DHX9, showed similar binding with both BAZ2A$_{WT}$ and BAZ2A$_{WY/GA}$ (Leone et al, 2017) (Fig 4B and Table S4). We identified 74 proteins that had decreased association with BAZ2A$_{WY/GA}$ compared with BAZ2A$_{WT}$. We applied the Search Tool for the Retrieval of Interacting Genes/Proteins database (STRING) (Szklarczyk et al, 2015) and found that proteins with decreased association with BAZ2A$_{WY/GA}$ were involved in pathways linked to the regulation of gene expression, chromosome organization, nuclear pore complex organization, and RNA processing (Fig 4C and Table S4). Among them, we found factors that regulate transcription and topological states of DNA such as topoisomerase 2A and 2B (TOP2A and TOP2B), histone demethylase KDM1A (also known as LSD1), and several components of cohesin protein complex (SMC1A, RAD21, STAG2). We also observed loss of BAZ2A interactions with factors involved in RNA processing and splicing (CTNNBL1, EFTUD2) and nuclear pore complex formation (NUP107, NUP133, NUP97). We validated the dependency of BAZ2A-TAM domain for BAZ2A interaction with TOP2A, SMC1A, RAD21, NUP107, and KDM1A by anti-HA IP of PC3 cells transfected with plasmids expressing F/H-mBAZ2A$_{WT}$, -mBAZ2A$_{WY/GA}$, and -mBAZ2A$_{\Delta TAM}$ followed by Western blot (Fig 4D). We confirmed these interactions also with endogenous BAZ2A by performing HA-IP in PC3 cells expressing endogenous BAZ2A tagged with HA and FLAG sequences (F/H-BAZ2A$_{end}$) (Fig 4E). To determine whether these BAZ2A interactions are mediated by the TAM domain itself or RNA bound to the TAM domain, we treated nuclear extracts of F/H-BAZ2A-PC3 cells and parental PC3 cells with RNase A and found that TOP2A and KDM1A association with BAZ2A was strongly reduced (Fig 4E). In contrast, BAZ2A association with RAD21, SMC1A, and NUP107 were not clearly affected by RNase A, suggesting that these interactions are BAZ2A-TAM-dependent but RNA independent. These results indicated that the BAZ2A-TAM domain mediates alone or in combination with RNA the interaction with proteins mainly involved in chromatin regulation. Furthermore, the data show that RNA may act as a scaffold for the association of BAZ2A with TOP2A and KDM1A (Fig 4F).

### KDM1A and TOP2A correlate with gene expression in PCa through the BAZ2A-TAM domain

Previous studies have implicated an involvement of both TOP2A and KDM1A in PCa. Elevated expression of KDM1A was shown to correlate with PCa recurrence (Kahl et al, 2006; Kashyap et al, 2013). Furthermore, high TOP2A levels were significantly associated with increased risk of systemic progression in PCa patients (Cheville et al, 2008). Consistent with these results, the analyses of two datasets comparing normal tissue, localized tumour, and metastatic PCa (Yu et al, 2004; Varambally et al, 2005; Chandran et al, 2007) revealed that *TOP2A* levels are highly elevated in metastatic PCa (Fig 5A and B). This pattern is very similar to *BAZ2A* expression

(*P*-values) for the experiments was calculated using two-tailed *t* test (*** < 0.001). **(G)** PCa subtypes and *PTEN* copy number alterations (can) in primary PCa groups defined by BAZ2A, TOP2A, and KDM1A expression levels. Quartile 1 (Q1): the top 25% of PCas with the highest expression. Quartile 4 (Q4): the top 25% of PCas with the lowest levels. The molecular subtypes of primary PCa (ERG fusions, ETV1/ETV4/FLI1 fusions or overexpression, and SPOP/FOXA1/IDH1 mutations) defined in Cancer Genome Atlas Research Network (2015) are shown.

levels that previous work also detected to be high in metastatic PCa relative to normal tissue and localized tumours (Gu et al, 2015). In contrast, the expression of *KDM1A* between metastatic PCa relative to localized tumour and normal tissue was less consistent between the two different datasets. However, the analysis of a large cohort of primary PCa and metastatic castration resistant PCa (CRPC) (333 and 118 tumours, respectively) (Cancer Genome Atlas Research Network, 2015; Robinson et al, 2015) revealed a significant positive correlation of both TOP2A and KDM1A levels with BAZ2A expression in both metastatic and primary tumors (Fig 5C–F). Furthermore, most of primary PCas with high levels of *TOP2A* or *KDM1A* belonged to the PCa subtype characterized by copy number alterations of the tumor suppressor gene *PTEN* and containing ERG fusion, which is the subtype characterizing PCa with high BAZ2A expression (Pietrzak et al, 2020) (Fig 5G). These results suggest that TOP2A and KDM1A might be functionally related to BAZ2A for the regulation of gene expression in PCa. To test this hypothesis, we analysed the expression of BAZ2A-TAM-repressed genes (i.e., up-regulated upon $BAZ2A_{\Delta TAM}$ expression in PC3 cells) and its connection to KDM1A and TOP2A using transcriptomic dataset from a large cohort of primary PCa (Cancer Genome Atlas Research Network, 2015). First, we classified tumours with high and low BAZ2A levels (83 primary PCas for each group, $BAZ2A^{high}$, the top 25% of PCas with the highest BAZ2A expression; $BAZ2A^{low}$, the top 25% of PCas with the lowest BAZ2A expression) and identified 409 genes that were significantly down-regulated in $BAZ2A^{high}$ compared with $BAZ2A^{low}$ tumours (Fig 6A and B). We found that 28% (113) of these genes were BAZ2A-TAM-repressed genes in PC3 cells. Importantly, 74% of these BAZ2A-TAM-dependent genes (84) were significantly down-regulated by comparing $BAZ2A^{high}/KDM1A^{high}$ tumours (i.e., 39 PCas that scored as the top 25% with the highest BAZ2A and KDM1A expressions, respectively) compared with $BAZ2A^{low}/KDM1A^{low}$ tumours (41 PCas) (Figs 6A and B and S3A). The rest of genes not showing statistical significance were also down-regulated upon treatment with KDM1Ai, except for two genes that were up-regulated (Fig 6B). We repeated a similar analysis for TOP2A and found that 90% (102) of genes repressed by BAZ2A-TAM in PC3 cells and down-regulated in primary $BAZ2A^{high}$ PCas were also down-regulated in $BAZ2A^{high}/TOP2A^{high}$ tumours compared with $BAZ2A^{low}/TOP2A^{low}$ (Figs 6C and D and S3B). Moreover, most of these BAZ2A-TAM-regulated genes were down-regulated in both $BAZ2A^{high}/KDM1A^{high}$ and $BAZ2A^{high}/TOP2A^{high}$ (Fig 6E and F). This is result is also consistent with the fact that about 50% of $BAZ2A^{high}/KDM1A^{high}$ and $BAZ2A^{high}/TOP2A^{high}$ corresponds to tumours with high TOP2A and KDM1A expression levels, respectively (Q1) (Fig S3C). Gene ontology terms of these genes correspond to metabolic and wound healing-related processes, which are very similar to the terms found for $BAZ2A_{\Delta TAM}$-regulated genes in PC3 cells (Fig S3D and Table S5). Next, we analysed a dataset comparing metastatic PCa with normal and primary tumours (Yu et al, 2004; Chandran et al, 2007). We found that a large fraction (44%) of these BAZ2A-TAM-regulated genes down-regulated in $BAZ2A^{high}/KDM1A^{high}$ or $BAZ2A^{high}/TOP2A^{high}$ primary PCa were significantly repressed in metastatic PCa compared with normal and primary tumours (44% and 47%, respectively), whereas only few were up-regulated (6% and 8%) (Fig 6G and H). Gene ontology terms of these genes correspond to developmental and wound healing-related processes (Fig S3E and Table S5). Taken

together, these results suggest an important link between BAZ2A-TAM domain, KDM1A, and TOP2A in the regulation of genes linked to PCa.

## Inhibition of KDM1A and TOP2A affects BAZ2A-TAM-repressed genes in PCa cells

To determine whether pharmacological inhibition of KDM1A and TOP2A affects the expression of genes regulated by the BAZ2A-TAM domain, we treated PC3 cells with the KDM1A inhibitor OG-L002 (KDM1Ai) or the TOP2A inhibitor ICR-193 (TOP2Ai) (Ishida et al, 1991) and performed RNA-seq analyses. In the case of KDM1A inhibition, we identified 453 up-regulated genes and 152 down-regulated genes in PC3 cells treated with KDM1Ai compared with control cells ($log_2$ fold change ±0.58, $P < 0.05$, Fig 7A and Table S6). Gene ontology term analysis revealed that the up-regulated genes are implicated in pathways linked to development, whereas down-regulated genes are linked to defence response processes (Fig 7B and Table S7). Next, we asked whether BAZ2A-TAM-regulated genes were also regulated by KDM1A. We found that 32% and 26% of genes, respectively, up-regulated or down-regulated by $BAZ2A_{\Delta TAM}$ expression were also significantly affected by KDM1Ai treatment (Figs 7C and S4A). Importantly, most of these BAZ2A-TAM-regulated genes affected by KDM1Ai treatment showed similar changes upon the expression of $BAZ2A_{\Delta TAM}$ and treatment with KDM1Ai. Genes up-regulated by $BAZ2A_{\Delta TAM}$ and KDM1Ai in PC3 cells included all the 84 BAZ2A-TAM-repressed genes that were down-regulated in $BAZ2A^{high}/KDM1A^{high}$ primary PCa (Figs 6A and 7D), suggesting an important role of KDM1A activity in BAZ2A-TAM-dependent gene regulation in PCa.

Next, we analysed the effect of TOP2A inhibition with TOP2Ai ICR-193. We identified 722 up-regulated genes and 410 down-regulated genes in PC3 cells treated with TOP2Ai compared with control cells ($log_2$ fold change ±0.58, $P < 0.05$, Fig 7E and Table S8). Gene ontology term analysis revealed that down-regulated genes were strongly implicated in cell cycle progress and chromosome segregation, an expected result given the known role of TOP2A in these processes (Chen et al, 2015) (Fig 7F and Table S9). We asked whether BAZ2A-TAM-regulated genes were also regulated by TOP2A and found that 52% and 35% of genes significantly up-regulated or down-regulated by $BAZ2A_{\Delta TAM}$ expression were also significantly affected by TOP2Ai treatment (Figs 7G and S4B). In the case of genes up-regulated by $BAZ2A_{\Delta TAM}$ in PC3 cells, 52% of them were significantly up-regulated by TOP2Ai treatment and included 52 out of the 102 BAZ2A-TAM-repressed genes that were down-regulated in $BAZ2A^{high}/TOP2A^{high}$ primary PCa (Figs 6C and 7H). Moreover, about half of these BAZ2A-TAM and TOP2Ai up-regulated genes were also significantly regulated by treatment with KDM1Ai and most of them were up-regulated (Fig 7I). We observed similar results for genes down-regulated by TOP2Ai (Fig S4B). These results further indicate that KDM1A and TOP2A are implicated in BAZ2A-TAM-regulated gene repression in PCa. Importantly, measurements of 45S pre-rRNA levels upon treatment with KDM1Ai or TOP2Ai revealed no significant changes, underscoring distinct mechanisms of BAZ2A repression in PCa cells based on the dependency of TOP2A and KDM1A (Fig 7J).

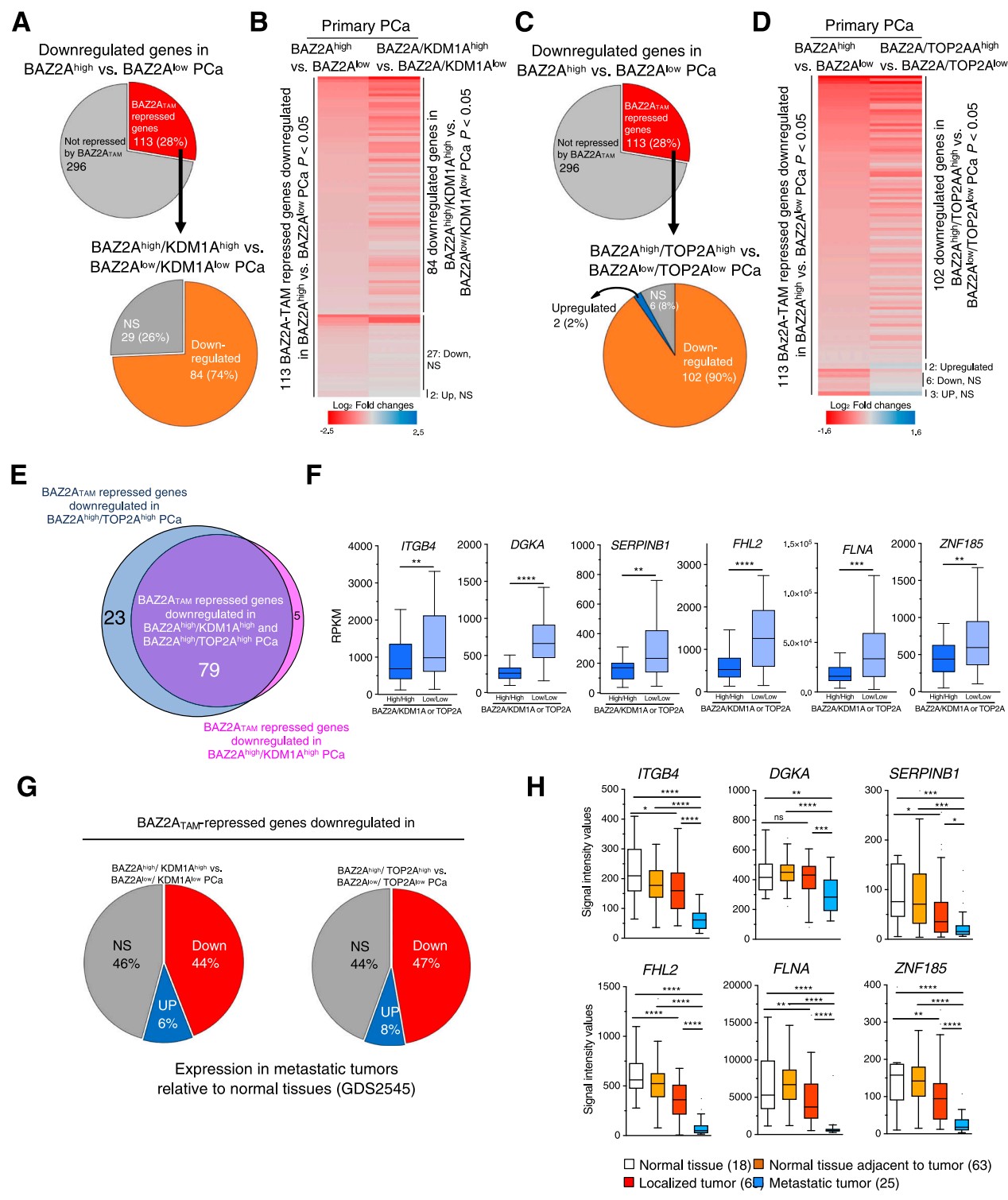

**Figure 6. Down-regulation of BAZ2A-TAM-repressed genes correlates with tumours expressing high levels of KDM1a and TOP2A.**
**(A)** Pie chart (top panel) showing the number of genes significantly down-regulated by comparing highly and low BAZ2A expressing primary PCas. The number of genes up-regulated upon the expression of BAZ2A$_{\Delta TAM}$ in PC3 cells is shown. The pie chart below shows the number of genes that are significantly down-regulated in BAZ2A$^{high}$/ KDM1A$^{high}$ tumours compared with BAZ2A$^{low}$/KDM1A$^{low}$ tumours. **(B)** Heatmap of log$_2$ fold change expression levels of BAZ2A-TA-repressed genes down-regulated in BAZ2A$^{high}$ versus BAZ2A$^{low}$ PCas and their corresponding log$_2$ fold change expression in BAZ2A$^{high}$/KDM1A$^{high}$ tumours compared with BAZ2A$^{low}$/KDM1A$^{low}$ tumours. **(C)** Pie chart (top panel) is the same as in 6A. The pie chart below shows the number of genes that are significantly down-regulated in BAZ2A$^{high}$/TOP2A$^{high}$ tumours compared with BAZ2A$^{low}$/TOP2A$^{low}$ tumours. **(D)** Heatmap of log$_2$ fold change expression levels of BAZ2A-TAM repressed genes down-regulated in BAZ2A$^{high}$ versus BAZ2A$^{low}$ PCa and their corresponding log$_2$ fold change expression in BAZ2A$^{high}$/TOP2A$^{high}$ tumours compared with BAZ2A$^{low}$/TOP2A$^{low}$ tumours. **(E)** Proportional Venn diagram showing BAZ2A$_{TAM}$-repressed genes down-regulated in both BAZ2A$^{high}$/KDM1A$^{high}$ and BAZ2A$^{high}$/TOP2A$^{high}$ primary PCas. **(F)** Box plots showing expression levels (RPKM) of

Finally, we asked how many BAZ2A-TAM-regulated genes that are located in the nearest linear proximity to BAZ2A-bound C2-enhancers were affected by the treatment of PC3 cells with KDM1Ai or TOP2Ai (Fig 8). More than one-third (39%) of genes up-regulated upon BAZ2A$_{\Delta TAM}$ expression were significantly affected by KDM1Ai and the majority (77%) show similar changes to BAZ2A$_{\Delta TAM}$ expression, being up-regulated compared with control cells (Fig 8A). These results indicate that KDM1A activity is implicated in the repression of genes having their corresponding C2-enhancers bound by BAZ2A and repressed through the BAZ2A-TAM domain. In the case of TOP2A inhibition, we found that 52% of genes up-regulated by BAZ2A$_{\Delta TAM}$ and TOP2Ai were also significantly up-regulated upon treatment with KDM1Ai and only 2% were down-regulated (Fig 8B). All these results further support a mechanism by which the BAZ2A-TAM domain regulates gene repression in PCa through its interaction with either TOP2A and KDM1A or both (Fig 8C).

## Discussion

Previous studies have implicated BAZ2A in aggressive PCa (Gu et al, 2015; Pietrzak et al, 2020; Pena-Hernandez et al, 2021); however, the mechanisms by which BAZ2A regulates gene expression in PCa cells remained elusive. In this work, we showed the importance of the BAZ2A-TAM domain and its interaction with RNA in repressing genes critical to PCa.

Studies in mouse non-cancer and differentiated cells showed that the BAZ2A-TAM domain associates with lncRNA pRNA that is required for BAZ2A targeting and silencing of rRNA genes (Mayer et al, 2006; Guetg et al, 2012; Anosova et al, 2015; Leone et al, 2017). Here, we show that pRNA mediates rRNA gene silencing in PCa cells, but it is not required for the repression of the other BAZ2A-regulated genes. Furthermore, we showed that the TAM domain is not required for BAZ2A binding with BAZ2A-regulated genes, whereas it is important for the interaction with rRNA genes as previously described (Mayer et al, 2006). These results further support a role of BAZ2A in PCa that goes beyond the known regulation of rRNA gene transcription (Santoro et al, 2002; Guetg et al, 2012; Gu et al, 2015) and suggest different mechanisms by which BAZ2A represses gene expression in PCa cells.

We showed that the BAZ2A-TAM domain serves for the association with factors implicated in the regulation of gene expression, chromatin organization, nuclear pore complex, and RNA splicing. Recent work demonstrated that BAZ2A recruitment to target loci, including a class of inactive (C2) enhancers, is mainly mediated by its bromodomain that specifically interacts with H3K14ac (Peña-Hernández et al, 2021). Our results indicate that BAZ2A regulation in PCa cells depends on both the bromo and the TAM domains; the

bromodomain acts as an epigenetic reader targeting BAZ2A to chromatin regions enriched in H3K14ac, whereas the TAM domain serves for the association with factors that impact gene expression (Fig 8C).

We showed that BAZ2A associates with RNA in PCa cells, an interaction that is mediated by the TAM domain. Our data suggested that RNA is required for the interaction of BAZ2A with TOP2A and the histone demethylase KDM1A and our future work will aim to identify these BAZ2A-interacting RNAs. On the other hand, the BAZ2A-TAM domain can also mediate protein interactions in an RNA-independent manner as in the case of cohesin components. Interestingly, the interaction of KDM1A with RNA has been previously reported by showing that the long intergenic noncoding RNAs (lincRNAs) HOTAIR serves as a scaffold that mediates the interaction of KDM1A and PRC2, thereby coordinating the targeting of histone H3K27me3 and H3K4 demethylation on target genes (Tsai et al, 2010). Thus, by analogy, BAZ2A might recruit KDM1A to target sites through a yet unknown RNA. The reported ability of KDM1A to demethylate mono- and di-methylated lysine 4 of histone H3 (Rudolph et al, 2013) can probably serve to keep BAZ2A-bound C2-enhancers in an inactive state and thus contributing to gene repression. Accordingly, treatment of PC3 cells with KDM1A inhibitors up-regulated the expression of genes repressed by the BAZ2A-TAM domain and located close to inactive BAZ2A-bound C2-enhancers that are characterized by low H3K4me1 and H3K27ac levels (Peña-Hernández et al, 2021). Moreover, our results showed that BAZ2A-TAM-repressed genes are frequently repressed in both primary and metastatic PCas with elevated expression of both BAZ2A and KDM1A or TOP2A. KDM1A is highly expressed in various human malignancies and its activities were linked to carcinogenesis, making it a possible target for anticancer treatments (Maiques-Diaz & Somervaille, 2016). In particular, KDM1A was shown to promote the survival of CRPC by activating the lethal PCa gene network and supporting the proliferation of PCa cells (Liang et al, 2017; Sehrawat et al, 2018). Furthermore, several inhibitors were recently shown to reduce the proliferative potential of PCa cells and PCa growth (Yang et al, 2017; Etani et al, 2019). Thus, the association of KDM1A with BAZ2A defined novel mechanisms by which KDM1A can act in PCa.

The other BAZ2A-TAM domain and RNA-dependent associating factor is TOP2A. As in the case of KDM2A, also, TOP2A has been reported to be frequently overexpressed in aggressive PCa and serves as an indicator of a poor outcome (Cheville et al, 2008; de Resende et al, 2013; Labbé et al, 2017). Furthermore, high levels of TOP2A were also found in CRPC (Hughes et al, 2006). Despite these correlations, however, the exact mechanism underlying the more aggressive phenotype associated with TOP2A is not known. Interestingly, BAZ2A–TOP2A interaction was previously detected in mouse embryonic stem cells (ESCs), where BAZ2A does not associate with rRNA genes and rRNA gene silencing is impaired

---

BAZ2A$_{TAM}$ repressed genes down-regulated in both BAZ2A$^{high}$/KDM1A$^{high}$ and BAZ2A$^{high}$/TOP2A$^{high}$ primary PCas. Error bars represent s.d. Statistical significance (*P*-values) for the experiments was calculated using two-tailed *t* test (** < 0.01; *** < 0.001; **** < 0.0001). **(G)** Pie charts showing the proportion of BAZ2A$_{TAM}$-repressed genes down-regulated in both BAZ2A$^{high}$/KDM1A$^{high}$ and BAZ2A$^{high}$/TOP2A$^{high}$ primary PCas that are up-regulated, down-regulated, and non-significantly changed (NS) in metastatic tumours compared with normal tissue. Dataset (GDS2545) are from Yu et al (2004); Chandran et al (2007). **(H)** Boxplots showing expression profiles of BAZ2A$_{TAM}$ repressed genes down-regulated in both BAZ2A$^{high}$/KDM1A$^{high}$ and BAZ2A$^{high}$/TOP2A$^{high}$ primary and metastatic PCas. Dataset (GDS2545) are from Yu et al (2004); Chandran et al (2007). Error bars represent s.d. Statistical significance (*P*-value) was calculated using two-tailed *t* test (* <0.05, **<0.01, *** < 0.001, **** < 0.0001); ns, not significant.

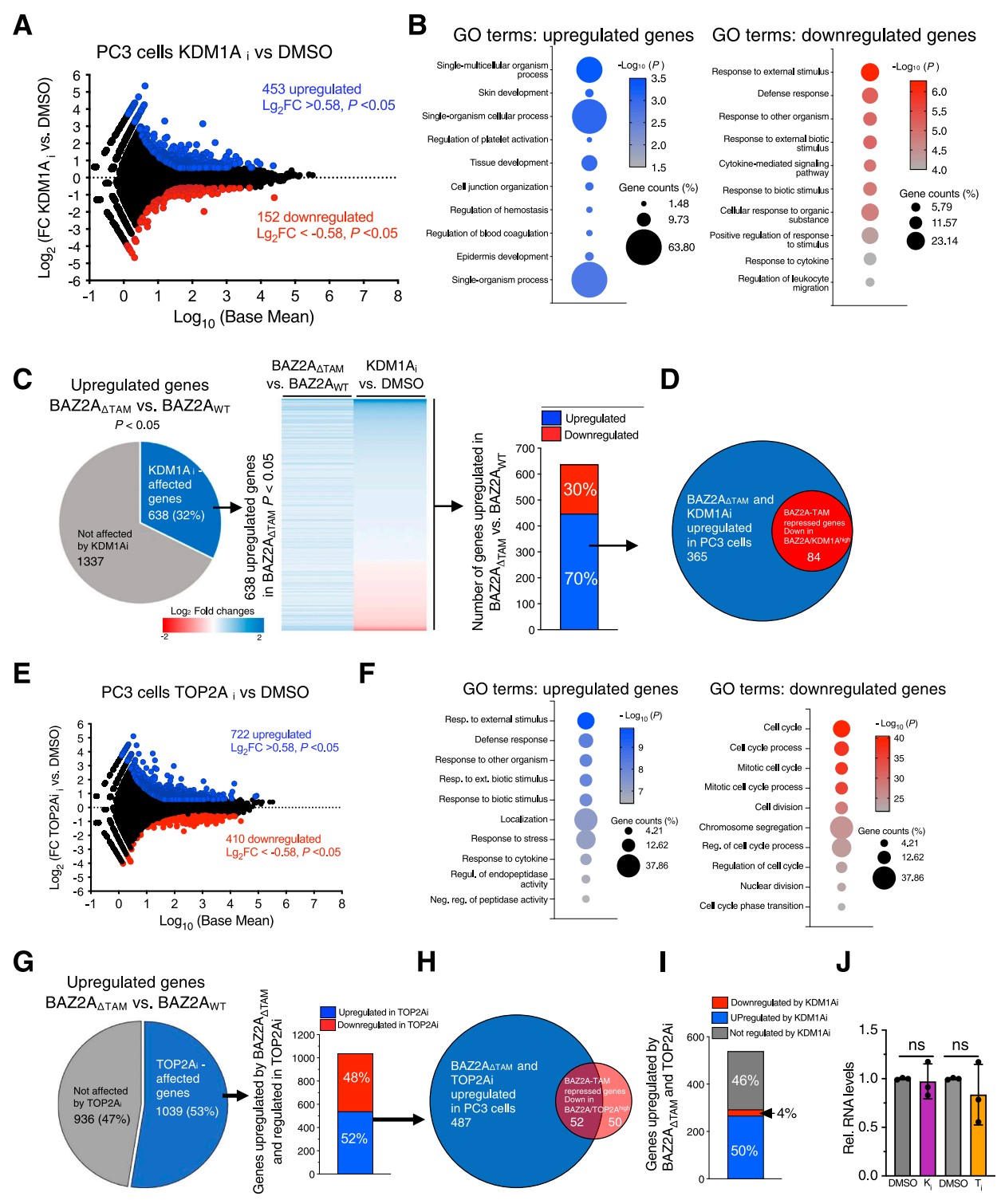

**Figure 7. KDM1A and TOP2A regulate BAZ2A-TAM-dependent genes in PCa cells.**
**(A)** KDM1A regulates the gene expression in PC3 cells. MA plot of the $\log_2$ fold change (FC) gene expression of PC3 cells treated with KDM1Ai OG-L002. Gene expression values of three replicates were averaged. Blue points represent up-regulated genes ($\log_2$FC > 0.58) and red points show down-regulated genes ($\log_2$FC < −0.58) and $P$-value < 0.05. **(B)** Top 10 gene ontology (GO) terms as determined using DAVID for genes up-regulated and down-regulated in PC3 cells treated with KDM1Ai. **(C)** Pie chart showing the number of genes significantly up-regulated upon BAZ2A$_{\Delta TAM}$ expression in PC3 cells that are affected upon KDM1Ai treatment. The middle panel shows the heat map of $\log_2$ fold change expression levels of genes up-regulated by BAZ2A$_{\Delta TAM}$ and affected by KDM1Ai. The right panel shows the number of BAZ2A-TAM-repressed and KDM1A-regulated genes that are significantly up- and down-regulated upon KDM1Ai treatment. **(D)** Proportional Venn diagram showing the number of up-regulated genes by BAZ2A$_{\Delta TAM}$ and KDM1Ai that are down-regulated in BAZ2A[high]/KDM1A[high] primary PCas. **(E)** TOP2A regulates gene expression in PC3 cells. MA plot of the $\log_2$ fold change (FC) gene expression of PC3 cells treated with TOP2Ai ICR-193. Gene expression values of three replicates were averaged. Blue points represent up-regulated

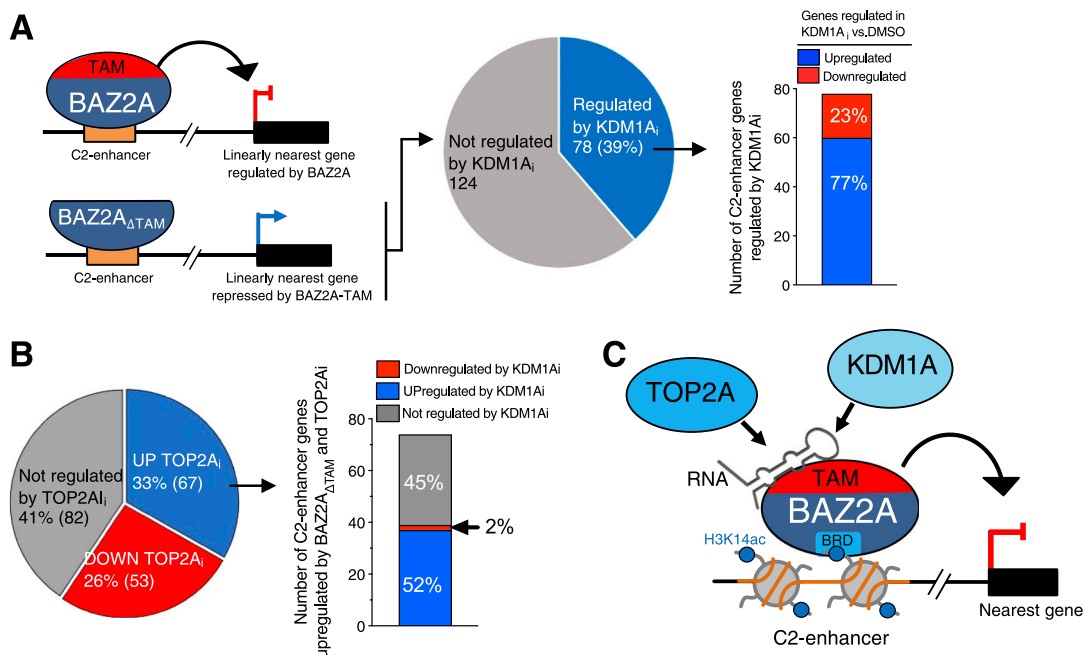

**Figure 8.  KDM1A and TOP2A repress genes that have their inactive C2-enhancers bound by BAZ2A.**
**(A)** A schema showing the up-regulation of genes in the linear nearest proximity to BAZ2A-bound C2-enhancers that are up-regulated upon BAZ2A$_{\Delta TAM}$ expression. The pie chart shows the number BAZ2A-TAM-dependent C2-enhancer genes that are significantly affected upon KDM1Ai treatment. In the right panel, it is shown that the number of BAZ2A-TAM-dependent C2-enhancer genes significantly up- and down-regulated upon KDM1Ai treatment. **(B)** The pie chart shows the number BAZ2A-TAM-dependent C2-enhancer genes that are significantly affected upon TOP2Ai treatment. In the right panel, it is shown the number of BAZ2A-TAM-dependent C2-enhancer genes up-regulated by TOP2Ai that are significantly up- and down-regulated upon KDM1Ai treatment. **(C)** Schema representing the role of BAZ2A-TAM, BAZ2A-bromodomain (BRD) and RNA in mediating targeting to C2-enhancers containing H3K14ac and the association with either TOP2A and KDM1A or both for the repression of gene expression in PCa cells.

([Dalcher et al, 2020](ref)). As in the case of PCa cells, BAZ2A and TOP2A in ESCs were shown to act together for the regulation of gene expression, suggesting that BAZ2A–TOP2A crosstalk is conserved among different species and cell context. In PCa, it has been reported that TOP2A cooperates with androgen receptor (AR) to induce transcription of target genes, suggesting that androgen ablation therapy might be less effective in the presence of high level of TOP2A protein ([Schaefer-Klein et al, 2015](ref)). In the case of BAZ2A, however, TOP2A activity seems to be related to gene repression, suggesting the TOP2A can act as activator or repressor depending on its interacting partners. The requirement of RNA for the association of TOP2A with BAZ2A suggests that TOP2A might be an RNA-binding protein. Consistent with this, recent high-resolution mapping of RNA-binding proteins detected a significant interaction of TOP2A with RNA ([He et al, 2016](ref); [Mullari et al, 2017](ref); [Perez-Perri et al, 2018](ref)). However, it remains to be investigated whether RNA in general or a specific RNA can mediate the association of TOP2A with distinct complexes, thereby specifying activating and/or repressing roles in gene expression. Finally, our data also revealed that tumours

expressing high BAZ2A levels also express high levels of KDM1A and TOP2A, suggesting common regulatory pathways. Indeed, a large portion of genes repressed through the BAZ2A-TAM domain could also be derepressed by pharmacological inhibition of both KDM1A and TOP2A, indicating that in some cases, TOP2A and KDM1A can act together in gene repression, although we cannot exclude cases where they act separately together with BAZ2A. In conclusion, our findings indicate that RNA-mediated interactions between BAZ2A and TOP2A and KDM1A are implicated in PCa and may prove to be useful for the stratification of PCa risk and treatment in patients.

# Materials and Methods

## Culture of PC3 cells

The PC3 cell line was purchased from the American Type Culture Collection. PC3 cells were cultured in RPMI 1640 medium and Ham's

genes (log$_2$FC > 0.58) and red points show down-regulated genes (log$_2$FC < −0.58) and *P*-value < 0.05. **(F)** Top 10 gene ontology (GO) terms as determined using DAVID for genes up-regulated and down-regulated in PC3 cells treated with TOP2Ai. **(G)** Pie chart showing the number of genes significantly up-regulated upon BAZ2A$_{\Delta TAM}$ expression in PC3 cells that are affected upon TOP2Ai treatment. Right panel shows the number of BAZ2A-TAM-repressed and TOP2A-regulated genes that are up- and down-regulated upon TOP2Ai treatment. **(H)** Proportional Venn diagram showing the number of up-regulated genes by BAZ2A$_{\Delta TAM}$ and TOP2Ai that are down-regulated in BAZ2A$^{high}$/TOP2A$^{high}$ primary PCas. **(I)** The number of genes up-regulated upon BAZ2A$_{\Delta TAM}$ expression and TOP2Ai treatment that are significantly up-regulated and down-regulated upon KDM1Ai treatment. **(J)** qRT–PCR showing 45S pre-rRNA levels upon treatment of PC3 cells with KDM1Ai or TOP2Ai. RNA levels were normalized to *GAPDH* mRNA and control cells treated with DMSO. Error bars represent SD of three independent samples. Statistical significance (*P*-value) was calculated using two-tailed *t* test. ns, not significant.

F12 medium (1:1; Gibco) containing 10% FBS (Gibco) and 1% penicillin–streptomycin (Gibco). $2.5 \times 10^6$ cells were seeded in 150 mm tissue culture dishes (TPP) and cultured for 2–3 d. All cells were regularly tested for mycoplasma contamination. The F/H-BAZ2A PC3 cell line was described in Peña-Hernández et al (2021).

## Construct design

cDNA corresponding to HA-FLAG-tagged mouse WT BAZ2A (F/H-BAZ2A$_{WT}$), CopGFP and shBAZ2A for human BAZ2A were cloned into DC-DON-SH01 vector (GeneCopoeia). Site-directed mutagenesis was performed to create BAZ2A mutants WY531/532GA (F/H-BAZ2A$_{WY/GA}$) and ΔTAM (HA/FLAG-BAZ2A$_{\Delta TAM}$). The sequencing of plasmids was performed to ensure the fidelity of sequences.

## Plasmid transfections

$1 \times 10^6$ of PC3 cells were seeded in a 100 mm culture dish (TPP), grown for 24 h, and transfected with 8 µg of plasmids and 8 µl of X-tremeGENE HP DNA Transfection Reagent (Roche). The cells were grown for 48 h posttransfection and collected for downstream analyses.

## RNA purification, reverse transcription, and quantitative PCR

RNA was purified with TRIzol reagent (Life Technologies). 1 µg total RNA was primed with random hexamers and reverse transcribed into cDNA using MultiScribe Reverse Transcriptase (Life Technologies). Amplification of samples without reverse transcriptase assured the absence of genomic or plasmid DNA (data not shown). The relative transcription levels were determined by normalization to *GAPDH or β-actin* mRNA levels, as indicated. qRT–PCR was performed with KAPA SYBR FAST (Sigma-Aldrich) on Rotor-Gene RG-3000 A (Corbett Research). Primer sequences are listed in Table S10.

## Treatment of PC3 cells with ICRF-193 and LG-001

$1 \times 10^6$ PC3 cells were seeded in a 100 mm culture dish (TPP), 24 h before the treatment. ICRF-193 (Sigma-Aldrich) or OG-L002 (MedChemExpress) was added directly to the medium to a final 10 µM concentration (Table S10). Cells were harvested for downstream analyses 48 h after treatment with the inhibitors.

## Chromatin immunoprecipitation

ChIP experiments were performed as previously described (Leone et al, 2017; Peña-Hernández et al, 2021). Briefly, 1% formaldehyde was added to cultured cells to cross-link proteins to DNA. Isolated nuclei were then digested with MNase (S7 Micrococcal nuclease; Roche) and briefly sonicated using a Bioruptor ultrasonic cell disruptor (Diagenode) to shear genomic DNA to an average fragment size of 200 bp. 200 µg of chromatin was diluted tenfold with ChIP buffer (16.7 mM Tris–HCl pH 8.1, 167 mM NaCl, 1.2 mM EDTA, 0.01% SDS, and 1.1% Triton X-100) and precleared for 2 h with 10 µl of packed Sepharose beads for at least 2 h at 4°C. Immunoprecipitation was performed overnight with the HA-magnetic beads (Pierce Anti-HA magnetic beads; Thermo Fisher Scientific). After

washing, elution, and reversion of cross-links, the eluates were treated with RNase A (1 µg). DNA was purified with phenol-chloroform, ethanol precipitated, and quantified by quantitative PCR. Primer sequences and antibodies are listed in Table S10.

## RNA-seq and data analysis

$1.5 \times 10^5$ PC3 cells were seeded into each well of a six-well culture dish (TPP), grown for 24 h, and transfected with constructs expressing either F/H-BAZ2A$_{WT}$ or F/H-BAZ2A$_{\Delta TAM}$ with simultaneous down-regulation of endogenous human BAZ2A. 4 µg of plasmid and 4 µl of X-tremeGENE HP DNA Transfection Reagent (Roche) were used per well. The cells were grown for 48 h posttransfection, collected by trypsinization, and pooled. Using FACS, the GFP(+) cells expressing F/H-BAZ2A$_{WT}$ or F/H-BAZ2A$_{\Delta TAM}$ were collected and their total RNA purified with TRIzol reagent (Life Technologies) as described above. Total RNA from three independent samples from PC3 cells expressing F/H-BAZ2A$_{WT}$ and from three independent samples expressing F/H-BAZ2A$_{\Delta TAM}$ were obtained. DNA contaminants were removed by treating RNA with 2U TURBO DNase I (Invitrogen) for 1 h at 37°C and the RNA samples were repurified using TRIzol. The quality of the isolated RNA was determined by Bioanalyzer 2100 (Agilent). Only those samples with a 260 nm/280 nm ratio between 1.8–2.1 and a 28S/18S ratio within 1.5–2 were further processed. The TruSeq RNA Sample Prep Kit v2 (Illumina, Inc.) was used in the succeeding steps. Briefly, total RNA samples (100–1,000 ng) were poly(A) enriched and then reverse transcribed into double-stranded cDNA. The cDNA samples were fragmented, end-repaired, and poly-adenylated before ligation of TruSeq adapters containing the index for multiplexing. Fragments containing TruSeq adapters on both ends were selectively enriched with PCR. The quality and quantity of the enriched libraries were validated using Qubit (1.0) Fluorometer and the Caliper GX LabChip GX (Caliper Life Sciences, Inc.). The product was a smear with an average fragment size of ~260 bp. The libraries were normalized to 10 nM in Tris-Cl 10 mM, pH 8.5 with 0.1% Tween 20. The TruSeq SR Cluster Kit HS4000 (Illumina, Inc) was used for cluster generation using 10 PM of pooled normalized libraries on the cBOT. Sequencing was performed on the Illumina HiSeq 2500 single-end 100 bp using the TruSeq SBS Kit HS2500 (Illumina, Inc). Reads were aligned to the reference genome (hg38) with Subread (i.e. subjunc, version 1.4.6-p4; [Liao et al, 2013]) allowing up to 16 alignments per read (options: –trim5 10 –trim3 15 -n 20 - m 5 -B 16 -H –allJunctions). Count tables were generated with Rcount (Schmid & Grossniklaus, 2015) and with an allocation distance of 10 bp for calculating the weights of the reads with multiple alignments, considering the strand information, and a minimal number of five hits. Variation in gene expression was analyzed with a general linear model in R with the package edgeR (version 3.12.0; [Robinson & Oshlack, 2010]). Genes differentially expressed between specific conditions were identified with linear contrasts using trended dispersion estimates and Benjamini–Hochberg multiple testing corrections. Genes with a *P*-value below 0.05 and a minimal fold change of 1.5 were considered as differentially expressed. These thresholds have previously been used characterizing chromatin remodeler functions (de Dieuleveult et al, 2016). Gene ontology analysis was performed with David Bioinformatics Resource 6.8 (Huang da et al, 2009).

## BAZ2A co-immunoprecipitation and mass spectrometric analysis

$1 \times 10^6$ PC3 cells were seeded in 100 mm culture dish (TPP), grown for 24 h, and transfected with 8 $\mu$g of plasmid DNA for expression of HA/FLAG-BAZ2A$_{WT}$, HA/FLAG-BAZ2A$_{WY/GA}$ or HA/FLAG-BAZ2A$_{\Delta TAM}$. Four culture dishes were used per condition. Cells were grown for another 48 h, harvested by scraping, and washed two times with PBS. The cells were then incubated for 10 min on ice in 3 ml of hypotonic buffer (10 mM HEPES pH 7.6, 1.5 mM MgCl$_2$, 10 mM KCl), spun down for 5 min, at 1,000$g$ at 4°C, supplemented with 0.5% Triton X-100, and incubated for another 10 min at 4°C. The obtained nuclei were spun down for 10 min, at 1,000$g$ at 4°C, resuspended in 500 $\mu$l of MNase digestion buffer (0.3 M Sucrose, 50 mM Tris–HCl pH 7.5, 30 mM KCl, 7.5 mM NaCl, 4 mM MgCl$_2$, 1 mM CaCl2, 0.125% NP-40, 0.25% NaDeoxycholate), and supplemented with 50 U of MNase (Roche). Samples were incubated at 30 min at 37°C under shaking. Next, the NaCl concentration was brought to 200 mM and samples were incubated for 10 min on ice. The samples were spun down at maximum speed, the pellets discarded, and 10% of the supernatant was collected as input samples. The remaining part of the supernatant was diluted four times with MNase buffer, supplemented with cOmpleteTM Protease Inhibitor Cocktail (Roche), and the samples were incubated overnight at 4°C with orbital shaking with 50 $\mu$l of HA magnetic beads (Thermo Fisher Scientific) that were prewashed three times with MNase buffer. The day after, the beads were washed three times for 30 min at 4°C with 1 ml of the wash buffer (20 mM HEPES pH 7.6, 20% glycerol, 200 mM NaCl, 1.5 mM MgCl$_2$, 0.2 mM EDTA, 0.02% NP-40). Half of the beads were further analyzed by Western blotting, whereas the other half was prepared for LC–MS/MS analyses. The dry beads were dissolved in 45 $\mu$l buffer containing 10 mM Tris + 2 mM CaCl$_2$, pH 8.2, and 5 $\mu$l of trypsin (100 ng/$\mu$l in 10 mM HCl) for digestion, which was carried out in a microwave instrument (Discover System; CEM) for 30 min at 5 W and 60°C. The samples were dried in a SpeedVac (Savant), dissolved in 0.1% formic acid (Romil), and an aliquot ranging from 5 to 25% was analyzed on a nanoAcquity UPLC (Waters Inc.) connected to a Q Exactive mass spectrometer (Thermo Fisher Scientific) equipped with a Digital PicoView source (NewObjective). Peptides were trapped on a Symmetry C18 trap column (5 $\mu$m, 180 $\mu$m × 20 mm; Waters Inc.) and separated on a BEH300 C18 column (1.7 $\mu$m, 75 $\mu$m × 150 m; Waters Inc.) at a flow rate of 250 nl/min using a gradient from 1% solvent B (0.1% formic acid in acetonitrile; Romil)/99% solvent A (0.1% formic acid in water; Romil) to 40% solvent B/60% solvent A within 90 min. Mass spectrometer settings were as follows: data-dependent analysis. Precursor scan range 350–1,500 m/z, resolution 70,000, maximum injection time 100 ms, threshold $3 \times 10^6$. Fragment ion scan range 200–2,000 m/z, resolution 35,000, maximum injection time 120 ms, threshold $1 \times 10^5$. Proteins were identified using the Mascot search engine (version 2.4.1; Matrix Science). Mascot was set up to search the SwissProt database assuming the digestion enzyme trypsin. Mascot was searched with a fragment ion mass tolerance of 0.030 D and a parent ion tolerance of 10.0 PPM. Oxidation of methionine was specified in Mascot as a variable modification. Scaffold (Proteome Software Inc.) was used to validate MS/MS-based peptide and protein identifications. Peptide identifications were accepted if they achieved a false discovery rate (FDR) of less than 0.1% by the Scaffold Local FDR algorithm. Protein identifications were accepted if they achieved an FDR of less than 1.0% and contained at least two identified peptides.

## BAZ2A co-immunoprecipitation with RNase A treatment

$2 \times 10^6$ of PC3 WT cells and $1 \times 10^6$ F/H-BAZ2A PC3 cells were seeded in a 150-mm culture dish (TPP) and grown for 48 h. Five culture dishes were used per condition. Cells were harvested by scraping and washed two times with PBS. The cells were incubated for 10 min on ice in 4 ml of hypotonic buffer (10 mM HEPES pH 7.6, 1.5 mM MgCl$_2$, 10 mM KCl), spun down for 5 min, at 1,000$g$ at 4°C, supplemented with 0.5% Triton X-100, and incubated for another 10 min at 4°C. The obtained nuclei were spun down for 10 min, at 1,000$g$ at 4°C, resuspended in 500 $\mu$l of MNase digestion buffer (0.3 M sucrose, 50 mM Tris–HCl pH 7.5, 30 mM KCl, 7.5 mM NaCl, 4 mM MgCl2, 1 mM CaCl2, 0.125% NP-40, 0.25% NaDeoxycholate) and supplemented with 40 U of MNase (Roche). The samples were shacken for 30 min at 37°C. Next, the NaCl concentration was brought to 200 mM and the samples were incubated for 10 min on ice. The samples were spun down at maximum speed. The pellets were discarded and 10% of the supernatant was collected as an input and further analysed by Western blot. The remaining part of the supernatant was diluted four times with MNase buffer, supplemented with cOmpleteTM Protease Inhibitor Cocktail (Roche), and 100 $\mu$g/ml RNase A (Thermo Fisher Scientific). The control samples were not treated with RNase A. The samples were incubated overnight at 4°C with orbital shaking with 25 $\mu$l of HA magnetic beads (Pierce Anti-HA magnetic beads; Thermo Fisher Scientific) prewashed three times with MNase buffer. The beads were washed three times for 30 min at 4°C with 1 ml of the wash buffer (20 mM HEPES pH 7.6, 20% glycerol, 200 mM NaCl, 1.5 mM MgCl$_2$, 0.2 mM EDTA, 0.02% NP-40) resuspended in a small volume of wash buffer. All the samples were loaded on a 6% home-made SDS–PAGE gel, run for 130 min at 120 V, and analysed further by Western blot using antibodies listed in Table S5.

# Data Availability

All raw data generated in this article using high throughput sequencing are accessible through National Center for Biotechnology Information Gene Expression Omnibus (GEO) database, https://www.ncbi.nlm.nih.gov/geo (accession no. GSE179743).

# Supplementary Information

# Acknowledgements

We thank Rostyslav Kuzyakiv for help in bioinformatics analyses. We thank Peter Hunziker, Catherine Aquino, and the Functional Genomic Center Zurich for assistance in sequencing and proteomic analysis. This work was supported by the Swiss National Science Foundation (31003A_173056 and 201268 to R Santoro), the National Center of Competence in Research RNA & Disease

(182880 funded by the SNSF), Forschungskredit of the University of Zurich (to M Roganowicz), Olga Mayenfisch Stifung, Krebsliga Zurich, Swiss Cancer Research Foundation (KFS-4527-08-2018-R; KFS-5488-02-2022), and an ERC grant (ERC-AdG-787074-NucleolusChromatin).

## Author Contributions

M Roganowicz: conceptualization, data curation, formal analysis, investigation, methodology, and writing—original draft.
D Bär: data curation, formal analysis, and investigation.
C Bersaglieri: resources, data curation, formal analysis, investigation, and writing—review and editing.
R Aprigliano: data curation, formal analysis, and investigation.
R Santoro: conceptualization, resources, data curation, supervision, funding acquisition, investigation, visualization, project administration, and writing—original draft, review, and editing.

## Conflict of Interest Statement

The authors declare that they have no conflict of interest.

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
