## [Reviewer comments · Life Science Alliance]

Life Science Alliance

BAZ2A-RNA mediated association with TOP2A and KDM1A represses genes implicated in prostate cancer

Marcin Roganowicz, Dominik Bär, Cristiana Bersaglieri, Rossana Aprigliano, and Raffaella Santoro
DOI: <https://doi.org/10.26508/lsa.202301950>

Corresponding author(s): Raffaella Santoro, University of Zurich

Review Timeline:

Submission Date:	2023-01-25
Editorial Decision:	2023-02-22
Revision Received:	2023-03-15
Editorial Decision:	2023-04-11
Revision Received:	2023-04-13
Accepted:	2023-04-17

Transaction Report:

February 22, 2023

Re: Life Science Alliance manuscript #LSA-2023-01950-T

Prof. Raffaella Santoro
University of Zurich
Molecular Mechanisms of Disease
Winterthurerstrasse 190
Zurich 8057
Switzerland

Dear Dr. Santoro,

Thank you for submitting your manuscript entitled "BAZ2A-RNA mediated association with TOP2A and KDM1A represses genes implicated in prostate cancer" to Life Science Alliance. The manuscript was assessed by an expert reviewer, whose comments are appended to this letter. We invite you to submit a revised manuscript addressing the Reviewer comments.

When submitting the revision, please include a letter addressing the reviewer comments point by point.

Thank you for this interesting contribution to Life Science Alliance. We are looking forward to receiving your revised manuscript.

Sincerely,

B. MANUSCRIPT ORGANIZATION AND FORMATTING:

Reviewer #1 (Comments to the Authors (Required)):

In this manuscript, Santoro and colleagues report a novel mechanism of BAZ2A function at C2-enhancers, which differs from its known role at rRNA genes. In the latter, BAZ2A is recruited to rRNA genes via interactions with TTF1, which are promoted by interactions of the BAZ2A-TAM domain with the long non-coding pRNA. At C2-enhancers, however, the BAZ2A-TAM domain mediates RNA-dependent interactions with TOP2A and KDM1A, which seem independent of the pRNA and unrelated to recruitment of BAZ2A to chromatin. These interactions lead to silencing of downstream genes. The authors find significant overlap between genes regulated by BAZ2A in a TAM-dependent fashion, and genes downregulated by TOP2A and KDM1A. A significant fraction of these genes are also downregulated in prostate cancer (PCa), where a high correlation between BAZ2A, TOP2A and KDM1A expression is detected. Altogether, this is a nice study showing a novel mechanism for BAZ2A with relevance for PCa. My main criticism is that the assays to assess RNA-binding by BAZ2A (and its TAM domain) in PCa cells are not convincing, thereby challenging an important message included in the manuscript title.

Major criticisms:

1) Figure 1A: The gels for Western blot and radioactive assessment are different, making the results difficult to interpret. First, if the gels are different one cannot serve as the loading control of the other. Second, in conditions of high RNase I-treatment, the radioactivity should collapse at the size of BAZ2A, and this is impossible to see if the gels used for Western-blot and radioactive detection are different. Typically, in these assays the radioactive nitrocellulose membrane is used for Western blot such that one can super-impose the bands. This has not been done here. Rather, there are two gels where the radioactive smear does not seem to coincide with the BAZ2A band in any condition. Please, show the radioactive signal associated to the exact same IP gel, and indicate on the autoradiography the exact size of H/F-BAZ2A.

2) Figure 1B: Same comment as for Figure 1A. In addition, here there is significant background from parental cells (not expressing the HA-tagged constructs), the radioactive smear is extensive and weak, and differences between wt and delta-TAM cells are minimal. These results are rather unconvincing, and any small difference should be corroborated by quantification of independent replicates. Furthermore, different BAZ2A RNA-defective mutants are used in the Western blot and the radioactive gels, which is not appropriate. Defective RNA-binding should be shown for both BAZ2A-WY/GA and BAZ2A-deltaTAM, as both proteins are extensively used in the manuscript and important conclusions are drawn as consequence of their defective RNA-binding.

3) Figure 1C-D: Please, describe the RNA control used in this experiment, and show its levels of expression compared to pRNA. Please, also explain the rationale to express these RNAs under the control of the rDNA promoter, which is regulated by BAZ2A.

4) Figure 2B: To observe equal expression of mBAZ2Awt and mBAZ2AdeltaTAM, the samples must have been run in the same gel. Is this the case? If so, please indicate in the figure legend that the line simply eliminates unwanted lanes from the same gel.

5) Figure 3F would require an additional replicate and statistic measures.

Minor criticisms:

6) H/F-BAZ2A-PC3 or F/H-BAZ2A-PC3? Please, use consistent nomenclature throughout the manuscript.

7) Please, include legends in Fig 2E (p-values, % genes?), as has been done for similar graphics in other figures.

8) Fig 3E and EV2: Why would genes associated to signal transduction and migration be downregulated in metastatic PCa? Isn't this counterintuitive?

9) Fig 4D: Please, show a better IP for Nup107. It is almost invisible in the IP of BAZ2Awt.

- 10) Figure 5A-B legend: Panels are separated by dataset, not by protein.
- 11) Page 10, lane 25: I failed to see the mentioned KDM1Ai-related data in Figure 6B.
- 12) Figure 6H is not mentioned in the main text.
- 13) Figure 8C: Should the RNA data be confirmed, the figure would benefit from drawing an RNA molecule helping to recruit TOP2A and KDM1A to BAZ2A.
- 14) Please, mention Figure 8B in page 12, lane 30.

Reviewer #1 (Comments to the Authors (Required)):

In this manuscript, Santoro and colleagues report a novel mechanism of BAZ2A function at C2-enhancers, which differs from its known role at rRNA genes. In the latter, BAZ2A is recruited to rRNA genes via interactions with TTF1, which are promoted by interactions of the BAZ2A-TAM domain with the long non-coding pRNA. At C2-enhancers, however, the BAZ2A-TAM domain mediates RNA-dependent interactions with TOP2A and KDM1A, which seem independent of the pRNA and unrelated to recruitment of BAZ2A to chromatin. These interactions lead to silencing of downstream genes. The authors find significant overlap between genes regulated by BAZ2A in a TAM-dependent fashion, and genes downregulated by TOP2A and KDM1A. A significant fraction of these genes are also downregulated in prostate cancer (PCa), where a high correlation between BAZ2A, TOP2A and KDM1A expression is detected. Altogether, this is a nice study showing a novel mechanism for BAZ2A with relevance for PCa. My main criticism is that the assays to assess RNA-binding by BAZ2A (and its TAM domain) in PCa cells are not convincing, thereby challenging an important message included in the manuscript title.

Author. We thank the reviewer for the positive comments. Here below, we clarified points that raised confusion. Changes in the text are highlighted in red.

Major criticisms:

Figure 1A: The gels for Western blot and radioactive assessment are different, making the results difficult to interpret. First, if the gels are different one cannot serve as the loading control of the other. Second, in conditions of high RNase I-treatment, the radioactivity should collapse at the size of BAZ2A, and this is impossible to see if the gels used for Western-blot and radioactive detection are different. Typically, in these assays the radioactive nitrocellulose membrane is used for Western blot such that one can super-impose the bands. This has not been done here. Rather, there are two gels where the radioactive smear does not seem to coincide with the BAZ2A band in any condition. Please, show the radioactive signal associated to the exact same IP gel, and indicate on the autoradiography the exact size of H/F-BAZ2A.

Author. We are sorry to have not provided sufficient details in the description of these experiments. We modified the text of the corresponding Figure legend, adding details that were not included in the previous version.

The RNA immunoprecipitations shown in Figs 1A and B have been performed using iCLIP protocols. The reason why we run two different gels for WB and RNA detection is that we wanted to isolate RNA-bound by BAZ2A transferred on the nitrocellulose membrane and generate libraries. Unfortunately, these libraries could not be used because of low quality. However, we think that the experiment is still valid and contains the correct controls. In iCLIP protocols, 8-10% of the IP is used for Western blot (WB) using PVDF membrane and the rest for radioactive RNA detection using nitrocellulose membrane that only allows the transfer of RNA and RNA-bound to proteins. The IP-WBs are an important control since they serve to show that the amount of BAZ2A-immunoprecipitates is the same in all indicated conditions. This important

information cannot be obtained by performing WB on the nitrocellulose membrane since only BAZ2A-RNA complexes can be transferred. Consequentially, BAZ2A-WB on the nitrocellulose membrane is not informative for the IP efficiency between samples. In trial experiments, we have also tried to perform WB on the radioactive nitrocellulose membrane, however, our antibodies did not work with the nitrocellulose membrane.

The size of BAZ2A on WB and the autoradiography cannot be directly compared since the samples were run with different electrophoresis gels according to published protocol (6% SDS-PAGE for WB and 3-8% NuPAGE TRIS-Acetate gels for the autoradiography). We included this information in the method section. Finally, we would also like to clarify that the experiments of Figures 1A and B do not represent the novelty of this work, but they were used to again introduce to a general readership (as done in the Introduction) that BAZ2A is an RNA binding protein. Indeed, BAZ2A interaction with RNA via the TAM domain has already been extensively described by many other studies, including structural studies. We modified the text of the Result section to make clear this point. Thus, the novelty of this work is not that BAZ2A interacts with RNA via TAM domain, but that BAZ2A-TAM domain mediates the interaction with TOP2A and KDM1A through RNA as demonstrated by the IP of endogenous BAZ2A upon treatment with RNase A (Fig. 4E).

2) Figure 1B: Same comment as for Figure 1A. In addition, here there is significant background from parental cells (not expressing the HA-tagged constructs), the radioactive smear is extensive and weak, and differences between wt and delta-TAM cells are minimal. These results are rather unconvincing, and any small difference should be corroborated by quantification of independent replicates. Furthermore, different BAZ2A RNA-defective mutants are used in the Western blot and the radioactive gels, which is not appropriate. Defective RNA-binding should be shown for both BAZ2A-WY/GA and BAZ2A-deltaTAM, as both proteins are extensively used in the manuscript and important conclusions are drawn as consequence of their defective RNA-binding.

Author. In this analysis, we used BAZ2A-WY/GA since, for reasons yet to clarify, at that time when we were making these experiments, we could not obtain similar expression levels between BAZ2A-WT and BAZ2A-delta-TAM as shown in other experiments of this manuscript. However, we think that the use of BAZ2A-WY/GA is reasonable since this mutant is well known to have impaired ability to bind to RNA as BAZ2A-deltaTAM (Mayer et al., 2006, Guetg et al., 2012). We modified the image of Fig. 1B using another exposure of the same gel which shows a lower RNA signal of BAZ2A-WY mutant and control relative to BAZ2A-WT. It is true that the quality of this image is low relative to the image of endogenous BAZ2A shown in Figure 1 and this could also depend on the fact that in this case we performed transient transfections. However, we think that the decrease of RNA signal of BAZ2A-WY mutant is clear. We also think that not showing RNA-BAZ2A-deltaTAM interaction analysis should not affect the results of this work since the TAM domain has been well characterized for its interaction with RNA, including structural studies (Anosova *et al*, 2015). In case the reviewer is not satisfied from this new image or our clarifications, we can delete this experiment from the manuscript since this does not represent the novelty of this work.

3) Figure 1C-D: Please, describe the RNA control used in this experiment, and show its levels of expression compared to pRNA. Please, also explain the rationale to express these RNAs under the control of the rDNA promoter, which is regulated by BAZ2A.

Author. The control RNA is a sequence previously shown to have low binding affinity to BAZ2A compared to pRNA (Savic et al. 2014; Guetg et al. 2012)). We included this information in the result section. We think that the measurements of the RNA control do not add any additional information since the data examined the role of pRNA in the expression of the indicated genes.

There are two major reasons why we placed pRNA under the control of the hrDNA promoter. First, in cells, pRNA is synthesized by RNA Pol I and this strategy has been employed in several other studies (Savic et al. 2014; Brenz et al., 2007). We included this information in the result section. The second reason is technical. Although ectopic expression of non-coding RNAs is usually done using RNA Pol III promoters, this cannot be done for pRNA since it contains a sequence that is recognized by Pol-III as a termination, thereby not allowing the entire transcription of pRNA.

4) Figure 2B: To observe equal expression of mBAZ2Awt and mBAZ2AdeltaTAM, the samples must have been run in the same gel. Is this the case? If so, please indicate in the figure legend that the line simply eliminates unwanted lanes from the same gel.

Author. Yes, the samples were run in the same gel. The original blots can be visualized in the original source data.

5) Figure 3F would require an additional replicate and statistic measures.

Author. We did not do a third replicate since both experiments showed that there is no difference in binding between BAZ2A-WT and BAZ2A-delta TAM on C2-enhancers. A third experiment would not have changed the result. The values of each experiment were marked with different colors (black and grey, respectively) and showed a similar trend.

Minor criticisms:

6) H/F-BAZ2A-PC3 or F/H-BAZ2A-PC3? Please, use consistent nomenclature throughout the manuscript.

Author. Thank you for having spotted this. We modified the text accordingly.

7) Please, include legends in Fig 2E (p-values, % genes?), as has been done for similar graphics in other figures.

Author. We included the legends in the corresponding Figure.

8) Fig 3E and EV2: Why would genes associated to signal transduction and migration be downregulated in metastatic PCa? Isn't this counterintuitive?

Author. The reviewer is right. By reanalyzing the data, we observed that the biggest GO term for these BAZ2A-TAM regulated genes (ca. 45%) is developmental processes, suggesting that BAZ2A promotes a dedifferentiation state that is a mark of aggressive PCa. We modified the text, accordingly.

9) Fig 4D: Please, show a better IP for Nup107. It is almost invisible in the IP of BAZ2Awt.

Author. Since Nup107 is not the major focus of this work, we deleted this panel while maintaining it in the co-IP with endogenous BAZ2A with RNase A treatment.

10) Figure 5A-B legend: Panels are separated by dataset, not by protein.

Author. We modified the legend, accordingly.

11) Page 10, lane 25: I failed to see the mentioned KDM1Ai-related data in Figure 6B.

Author. We have now included also Fig. 6B.

12) Figure 6H is not mentioned in the main text.

Author. We have now included Fig. 6H.

13) Figure 8C: Should the RNA data be confirmed, the figure would benefit from drawing an RNA molecule helping to recruit TOP2A and KDM1A to BAZ2A.

Author. Thank you for having notice it! We lost the RNA during the assembly of the Figure.

14) Please, mention Figure 8B in page 12, lane 30

Author. We included the citation of the indicated Figure.

April 11, 2023

RE: Life Science Alliance Manuscript #LSA-2023-01950-TR

Prof. Raffaella Santoro
University of Zurich
Molecular Mechanisms of Disease
Winterthurerstrasse 190
Zurich 8057
Switzerland

Dear Dr. Santoro,

Thank you for submitting your revised manuscript entitled "BAZ2A-RNA mediated association with TOP2A and KDM1A represses genes implicated in prostate cancer". We would be happy to publish your paper in Life Science Alliance pending final revisions necessary to meet our formatting guidelines.

- please address the Reviewer's remaining points
- please consult our manuscript preparation guidelines <https://www.life-science-alliance.org/manuscript-prep> and make sure your manuscript sections are in the correct order
- GEO accession GSE179743 should be made publicly accessible at this time

Figure Check:

-in Figure 2B, please make the black vertical line a bit thicker, and mention in the figure legend that the line indicates that the blot is not continuous

A. FINAL FILES:

B. MANUSCRIPT ORGANIZATION AND FORMATTING:

Sincerely,

Reviewer #1 (Comments to the Authors (Required)):

The manuscript by Santoro and colleagues has improved with the new text additions and corrections, but there are two points of disagreement:

1) Figures 1A-B: The major criticism regarding these figures remains. I understand that these figures were originally intended for iCLIP, but in the context of this manuscript they do not serve the purpose of demonstrating RNA-binding with sufficient quality. We routinely perform iCLIP experiments in my lab and, although it is correct that one does not perform Western-blot on the radioactive membrane from which the RNA is to be isolated, a prior optimization experiment must include a Western-blot on the radioactive membrane in order to precisely map the specific RNA-protein complexes that one must isolate later on. It is incorrect that nitrocellulose membranes only allow transfer of RNA and RNA-protein complexes. Proteins alone are also transferred. In fact, Western blots were performed on nitrocellulose membranes long time ago, before PVDF membranes were available. The authors mention that their antibodies did not work on nitrocellulose membranes, which questions again the quality of the experiment. As RNA-binding by the TAM domain of BAZ2A has been previously reported, including the relevance of the two amino acids that are being mutated, I suggest to just remove these figure panels from the manuscript.

2) Figure 1D: The authors mention that 'the measurements of the RNA control do not add any additional information since the data examined the role of pRNA in the expression of the indicated genes'. But this is being measured relative to the RNA control and, therefore, it is important to ensure that pRNA and the RNA control are expressed at similar levels. Please, do perform RT-qPCR to evaluate such relative levels.

Other than this, the manuscript is a nice study showing a novel mechanism for BAZ2A function and deserves publication.

Reviewer #1

The manuscript by Santoro and colleagues has improved with the new text additions and corrections, but there are two points of disagreement: 1) Figures 1A-B: The major criticism regarding these figures remains. I understand that these figures were originally intended for iCLIP, but in the context of this manuscript they do not serve the purpose of demonstrating RNA-binding with sufficient quality. We routinely perform iCLIP experiments in my lab and, although it is correct that one does not perform Western-blot on the radioactive membrane from which the RNA is to be isolated, a prior optimization experiment must include a Western-blot on the radioactive membrane in order to precisely map the specific RNA-protein complexes that one must isolate later on. It is incorrect that nitrocellulose membranes only allow transfer of RNA and RNA-protein complexes. Proteins alone are also transferred. In fact, Western blots were performed on nitrocellulose membranes long time ago, before PVDF membranes were available. The authors mention that their antibodies did not work on nitrocellulose membranes, which questions again the quality of the experiment. As RNA-binding by the TAM domain of BAZ2A has been previously reported, including the relevance of the two amino acids that are being mutated, I suggest to just remove these figure panels from the manuscript.

Author: We removed from the manuscript Figs. 1A&B and adjusted the text accordingly.

2) Figure 1D: The authors mention that 'the measurements of the RNA control do not add any additional information since the data examined the role of pRNA in the expression of the indicated genes'. But this is being measured relative to the RNA control and, therefore, it is important to ensure that pRNA and the RNA control are expressed at similar levels. Please, do perform RT-qPCR to evaluate such relative levels.

Author: We would like to better clarify our reasons why assessing equal levels of RNA-Control and pRNA does not add any additional information to an experiment aimed to determine whether pRNA could affect the expression of BAZ2A-target genes in PCa cells as in the case of rRNA genes.

The RNA-Control, which is an artificial sequence and is not present in the human or mouse transcriptomes or genomes, does not associate with BAZ2A and that is why has been used as control RNA in this experiment (now Fig. 1B). It would have been important to know the expression levels of RNA-Control in case of *in vitro* experiments to prove that BAZ2A specifically associates with BAZ2A, as done in our previous works (i.e., Mayer et al., Mol Cell). Further, it would also be important to make these measurements if the RNA-Control was a mutant pRNA, but this was not the experiment. Finally, we cannot generate relative levels of pRNA and Control-RNA. To make this comparison, we need to generate absolute values, and this is not easy since qRT-PCR usually generates relative values of the same RNA and not absolute values of different RNAs due to the use of different primers pairs. In addition, measurements of absolute RNA values by qPCR requires standard curves, usually genomic DNA or total RNA. However, this cannot be done for RNA-Control since this is an artificial sequence. Although we would like very much to please the Reviewer also for this point, we regret to not be able to provide the information that RNA-Control and pRNA are expressed at the same level since this will mean to repeat this complex experiment and establish qPCR measurements of absolute RNA values only to obtain an information that we think it is not relevant for the experiment shown in Fig. 1B.

Other than this, the manuscript is a nice study showing a novel mechanism for BAZ2A function and deserves publication.

Author: We thank the Reviewer for the positive comments on our work.

April 17, 2023

RE: Life Science Alliance Manuscript #LSA-2023-01950-TRR

Prof. Raffaella Santoro
University of Zurich
Molecular Mechanisms of Disease
Winterthurerstrasse 190
Zurich 8057
Switzerland

Dear Dr. Santoro,

Thank you for submitting your Research Article entitled "BAZ2A-RNA mediated association with TOP2A and KDM1A represses genes implicated in prostate cancer". It is a pleasure to let you know that your manuscript is now accepted for publication in Life Science Alliance. Congratulations on this interesting work.

DISTRIBUTION OF MATERIALS:

Again, congratulations on a very nice paper. I hope you found the review process to be constructive and are pleased with how the manuscript was handled editorially. We look forward to future exciting submissions from your lab.

Sincerely,
